# Intracristal space proteome mapping using super-resolution proximity labeling with isotope-coded probes

Myeong-Gyun Kang [1,10], Sanghee Shin [2,3,4,10], Dong-Gi Jang[2,5], Ohyeon Kwon[6], Song-Yi Lee[1,7,8], Pratyush Kumar Mishra [1], Minkyo Jung [9], Ji Young Mun [9], Jung-Min Kee [6] ✉, Jong-Seo Kim [2,5] ✉ & Hyun-Woo Rhee [1,2] ✉

Proximity labeling with engineered ascorbate peroxidase (APEX) has been widely used to identify proteomes within various membrane-enclosed sub-cellular organelles. However, constructing protein distribution maps between two non-partitioned proximal spaces remains challenging with the current proximity labeling tools. Here, we introduce a proximity labeling approach using isotope-coded phenol probes for APEX labeling (ICAX) that enables the quantitative analysis of the spatial proteome at nanometer resolution between two distinctly localized APEX enzymes. Using this technique, we identify the spatial proteomic architecture of the mitochondrial intracristal space (ICS), which is not physically separated from the peripheral space. ICAX analysis further reveals unexpected dynamics of the mitochondrial spatiome under mitochondrial contact site and cristae organizing system (MICOS) complex inhibition and mitochondrial uncoupling, respectively. Overall, these findings highlight the importance of ICS for mitochondrial quality control under dynamic stress conditions.

Biological components of cells localize to specific subcellular positions to perform their unique function. Profiling the subcellular localization and function of spatially distributed biomolecules offers comprehensive insights into the various cellular processes. Recently, proximity labeling (PL) methods such as engineered ascorbate peroxidase (APEX)[1,2], promiscuous biotin ligase (BioID/TurboID)[3–5], and photocatalyst (Micromap)[6,7] have been developed to identify numerous spatially resolved biological components. In situ-generated reactive molecules by PL enzymes or photocatalysts can undergo covalent conjugation with spatially localized biomolecules or the spatiome[8] within a reactive diffusion radius that depends on

their lifetime under physiological conditions. Currently, carbene (Micromap) and biotin-AMP (BioID/TurboID) are estimated to have a short labeling radius (few nanometers)[6,8], which is preferred for the identification of ligand binding pockets or direct interaction partners. By contrast, peroxidase-based labeling with biotin-phenoxyl radicals (APEX or HRP) shows a rather comprehensive labeling pattern (~200 nm radius)[6], which facilitates covering of the whole luminal, membrane-partitioned proteome of organelles such as the nucleus[9], mitochondrial matrix[2,10], and endoplasmic reticulum[11]. However, mapping the local proteome at the sub-compartment level in high resolution remains challenging.

[1]Department of Chemistry, Seoul National University, Seoul, Korea. [2]School of Biological Sciences, Seoul National University, Seoul, Korea. [3]The Research Institute of Basic Science, Seoul National University, Seoul, Korea. [4]Department of Cancer Biology, Dana-Farber Cancer Institute, Boston, MA, USA. [5]Center for RNA Research, Institute for Basic Science, Seoul, Korea. [6]Department of Chemistry, Ulsan National Institute of Science and Technology (UNIST), Ulsan, Korea. [7]Department of New Biology, Daegu Gyeongbuk Institute of Science & Technology (DGIST), Daegu, Korea. [8]New Biology Research Center, Daegu Gyeongbuk Institute of Science & Technology (DGIST), Daegu, Korea. [9]Neural Circuit Research Group, Korea Brain Research Institute, Daegu, Republic of Korea. [10]These authors contributed equally: Myeong-Gyun Kang, Sanghee Shin. ✉e-mail: jmkee@unist.ac.kr; jongseokim@snu.ac.kr; rheehw@snu.ac.kr

Intermembrane space (IMS) of mitochondria, enclosed by the outer mitochondrial membrane (OMM) and the inner mitochondrial membrane (IMM), plays a crucial role in various biological functions including ATP synthesis, apoptosis regulation, and the transport of metabolites, lipids, and proteins[12,13]. IMS can be spatially divided into distinct sub-domains, peripheral space and intracristal space (ICS), which allow for the functional separation of critical processes[13]. The ICS is surrounded by the cristae membrane and is suggested to be enriched with oxidative phosphorylation (OXPHOS) complexes for constituting the primary site of energy generation[13,14]. However, the ICS proteome remains poorly characterized, as no method currently exists to isolate such membraneless sub-compartments. Although the IMS proteome has been cataloged using APEX2 conjugated with IMS-targeting sequences[15], sub-IMS proteome mapping in the ICS remains challenging due to the lack of biochemical tools to identify ICS marker proteins. Thus, it is necessary to develop new PL tools that can definitively reveal the sub-IMS proteome by identifying ICS marker proteins.

Liquid chromatography-mass spectrometry (LC-MS)-based protein quantification, coupled with PL studies, provides critical insights into the identification of biotinylated proteins and their dynamic behavior within the cellular environment. Recently, we developed super-resolution proximity labeling techniques that enable the direct detection and quantification of biotin- or desthiobiotin-phenol (DBP)-labeled proteins at single-modified-peptide resolution using BioID/TurboID[16,17] and APEX2[10,11,18], respectively. From this analysis, we found that "sub-compartment-level" proteome distribution can be determined by comparing mass signal intensities of DBP-labeled peptides from multiple proteins of interest (POI)-APEX2 samples (see www.mitoatlas.org) using label-free quantification (LFQ). While the LFQ approach offers the advantage of comparative analysis among various samples, it requires stringent control to ensure that liquid chromatography and ionization results remain reliable and reproducible across all sets of samples, which is technically challenging.

To overcome this limitation, stable isotope-labeled amino acid cell culturing (SILAC)[19], tandem mass tag (TMT)[20], and isotope-coded affinity tag (ICAT)[21] can be employed. However, SILAC cannot be applied to non-dividing cells or human tissues; additionally, its application is hampered by incomplete and cost-ineffective incorporation of isotopes in animal models[22]. Chemical conjugation of TMTs with non-abundant PL-modified peptides may not be efficient and requires additional steps. ICATs, initially designed for in vitro labeling, are unsuitable for intracellular labeling because of their membrane-impermeable PEG linker (Fig. 1a).

In this study, we develop a duplexed super-resolution proximity labeling strategy using novel isotope-coded phenol probes for APEX labeling (ICAX). Our method enables more accurate and reproducible determination of the relative proximity of labeled proteins from two regional bait proteins, thereby overcoming the aforementioned limitations in live-cell applications. Using this approach, we identify proteins specifically localized in the ICS, and the ICS proteome is further mapped through ICAX analysis.

## Results

### Chemoenzymatic preparation of isotope-coded APEX probes

Since PL methods employ different labeling mechanisms, we compared Micromap, TurboID, and APEX2 to develop a quantitative analysis platform using isotope-coded probes. Micromap utilizes an iridium catalyst to generate a carbene intermediate that reacts with various amino acids by inserting into C-H bonds[23,24]. While Micromap is effective for identifying protein interacting partners, the short labeling radius (-10 nm) of the carbene[6] limits its utility for sub-compartment-level proteome mapping.

In contrast, TurboID has comparatively longer labeling radius (-35 nm)[25]; however, it exhibited high reactivity with endogenous biotin present under standard culture conditions (Supplementary

Fig. 1a–d), leading to substantial background from endogenous sources. Given these limitations, we selected APEX2 as the labeling enzyme for the development of our quantitative PL platform.

The stable isotope-coded phenol probe for APEX-based PL was developed using stable carbon isotopes ($^{13}$C) within the linker and phenol regions, rather than the commonly used deuterium, to avoid its known chromatographic lagging effect[26]. Given these design constraints, we selected a heavy tyramine moiety with $^{13}$C- or $^{15}$N-isotopes that could be readily prepared from the corresponding heavy tyrosine in a single step using the *Streptococcus faecalis* tyrosine decarboxylase (Fig. 1a)[27]. Diverse forms of $^{13}$C- or $^{15}$N-labeled tyrosine isotopologues are commercially available, enabling flexibility and expandability of the isotope-coding strategy. Using a heavy tyramine ($^{13}C_8$, $^{15}N_1$- tyramine), we synthesized heavy desthiobiotin-phenol (HDBP, 343 Da) via coupling with desthiobiotinyl-NHS, resulting in the formation of a probe that is 9 Da heavier than the light desthiobiotin-phenol probe (LDBP, 334 Da) (Fig. 1a). Proximal proteins can be labeled using HDBP and LDBP, with two distinctly localized APEX proteins. The distance between the labeled proteins was estimated based on the mass signal intensity ratio between the APEX proteins (Fig. 1b).

To validate the isotopic purity of the probes, we performed LC-MS analysis of LDBP and HDBP individually and confirmed minimal cross-contamination (Supplementary Fig. 2a). Additional labeling experiments in mitochondrial targeting sequence (MTS)-APEX2-expressing cells (Supplementary Fig. 2b–c) showed negligible isotopic crossover, with only <0.6% of the peptides being detected in the paired channel (Supplementary Fig. 2d). These values are within the expected range given the 1% FDR threshold applied during peptide identification and do not affect the accuracy of ICAX quantification.

To further confirm whether LDBP and HDBP exhibit equivalent labeling efficiency in a cellular context, we conducted an in vivo benchmarking experiment using MTS-APEX2 as a model system for proximity labeling. Cells were treated with an equimolar pre-mixed solution of LDBP and HDBP to ensure simultaneous and competitive access of both probes to proximal protein targets under identical enzymatic and cellular conditions (Fig. 1c). This design enables a direct, internal comparison of the labeling efficiency across isotopic channels, independently of potential biological or technical variation. Quantitative comparison of paired signals demonstrated nearly identical intensities between LDBP- and HDBP-labeled peptides, with high reproducibility across replicates (Fig. 1d and Supplementary Data 3), confirming equivalent labeling efficiency of the two probes under physiological conditions.

Next, we conducted experiments using ICAX, wherein MTS-APEX2 cells were labeled with HDBP and LDBP, individually. The HDBP- and LDBP-labeled proteins were mixed with the lysates, and LC-MS/MS analysis was performed after enrichment of the labeled peptides (Fig. 1e). Direct detection of biotinylated peptides using LC-MS/MS produced spectra with clear isotopic peaks corresponding to LDBP- and HDBP- labeled species (Fig. 1f). A log2-transformed intensity scatter plot exhibited high accuracy and reproducibility of ICAX analysis using HDBP and LDBP (slope = 0.989 ± 0.003, $R^2$ = 0.976 ± 0.003, mean ± SEM; Fig. 1g and Supplementary Fig. 3a), which outperformed the LFQ approach with LDBP (slope = 0.956 ± 0.016, $R^2$ = 0.930 ± 0.005, mean ± SEM) and HDBP (slope = 0.952 ± 0.006, $R^2$ = 0.926 ± 0.008, mean ± SEM; Supplementary Fig. 3b and Supplementary Data 4).

Since light biotin-phenol (LBP) has been most widely used in APEX-based proximity labeling, we performed the same experiments with initially synthesized heavy biotin-phenol (HBP) and LBP probes for comparison (Supplementary Fig. 4a). However, these probes showed lower accuracy in probe mix (Supplementary Fig. 4b–f) and ICAX experiments (slope = 0.980 ± 0.001, $R^2$ = 0.962 ± 0.006, mean ± SEM; Supplementary Fig. 4g) owing to inefficient elution compared with HDBP and LDBP[18].

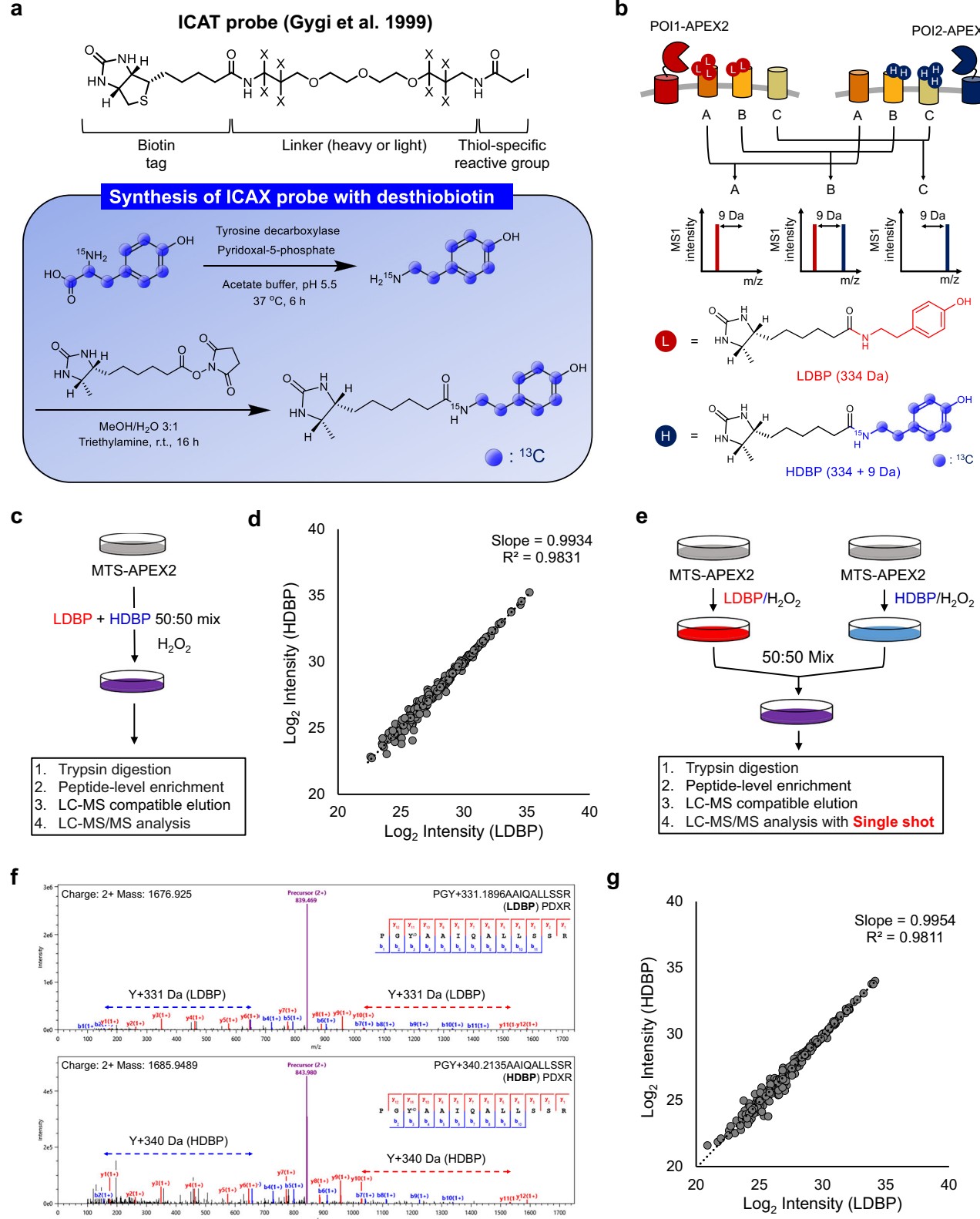

## Identification of TMEM177 specifically localized in the ICS

We conducted an ICAX experiment with HDBP and LDBP to resolve the two spatially distinct APEX2-labeled proteomes in the mitochondrial IMS. In our experiment, we utilized IMM-targeted APEX2 and OMM-targeted APEX2 to identify the vertically ordered proteome distribution between the IMM (boundary membrane and cristae membrane) and OMM[28]. Additionally, OMM porosity[15] allowed DBP radical to diffuse across it, which enabled our ICAX analysis to generate differential mass intensities of IMS-localized proteins based on distance. This approach has high potential for mapping proteins in the ICS, a sub-compartment surrounded by cristae membranes[29], which is more likely to be labeled by IMM-APEX2 rather than OMM-APEX2 (Fig. 2a).

We used two stable cell lines expressing IMM-targeted APEX2 (SCO1-APEX2) and OMM-targeted APEX2 (TDRKH-APEX2). In the IMM-

**Fig. 1 | Chemical synthesis of isotope-coded ascorbate peroxidase probes for quantitative proximity labeling. a** Chemical structure of heavy isotope-coded ascorbate peroxidase (APEX) probes used in this study. APEX probes were synthesized from tyrosine and conjugated with a desthiobiotin moiety. **b** Schematic representation of the quantitative mass analysis of proximity-dependent, labeled tyrosine residues using APEX probes. **c** Schematic representation of the experimental design of the equal activity test of desthiobiotin conjugated APEX probes (light desthiobiotin-phenol [LDBP] and heavy desthiobiotin-phenol [HDBP], 50:50 ratio mixture) on cells expressing mitochondrial targeting sequence (MTS)-APEX.

**d** Scatter plot showing MS1 intensity of both LDBP- and HDBP-modified proteins labeled using MTS-APEX2 with equimolar pre-mixed probe solutions (*n* = 3 biological replicates). R-squared ($R^2$) values and trendline slopes were used to validate accuracy. **e** Experimental procedure of duplexed quantification using the ICAX method. **f** Annotated mass spectrometry (MS)/MS spectra of the LDBP- and HDBP-modified peptides (PGY*AAIQAALSSR) for the PDXR protein. **g** Scatter plot showing the MS1 intensity of both LDBP- and HDBP-modified peptides identified by ICAX analysis. $R^2$ values and trendline slopes in biological triplicate experiments are listed in the Supplementary Fig. 3a.

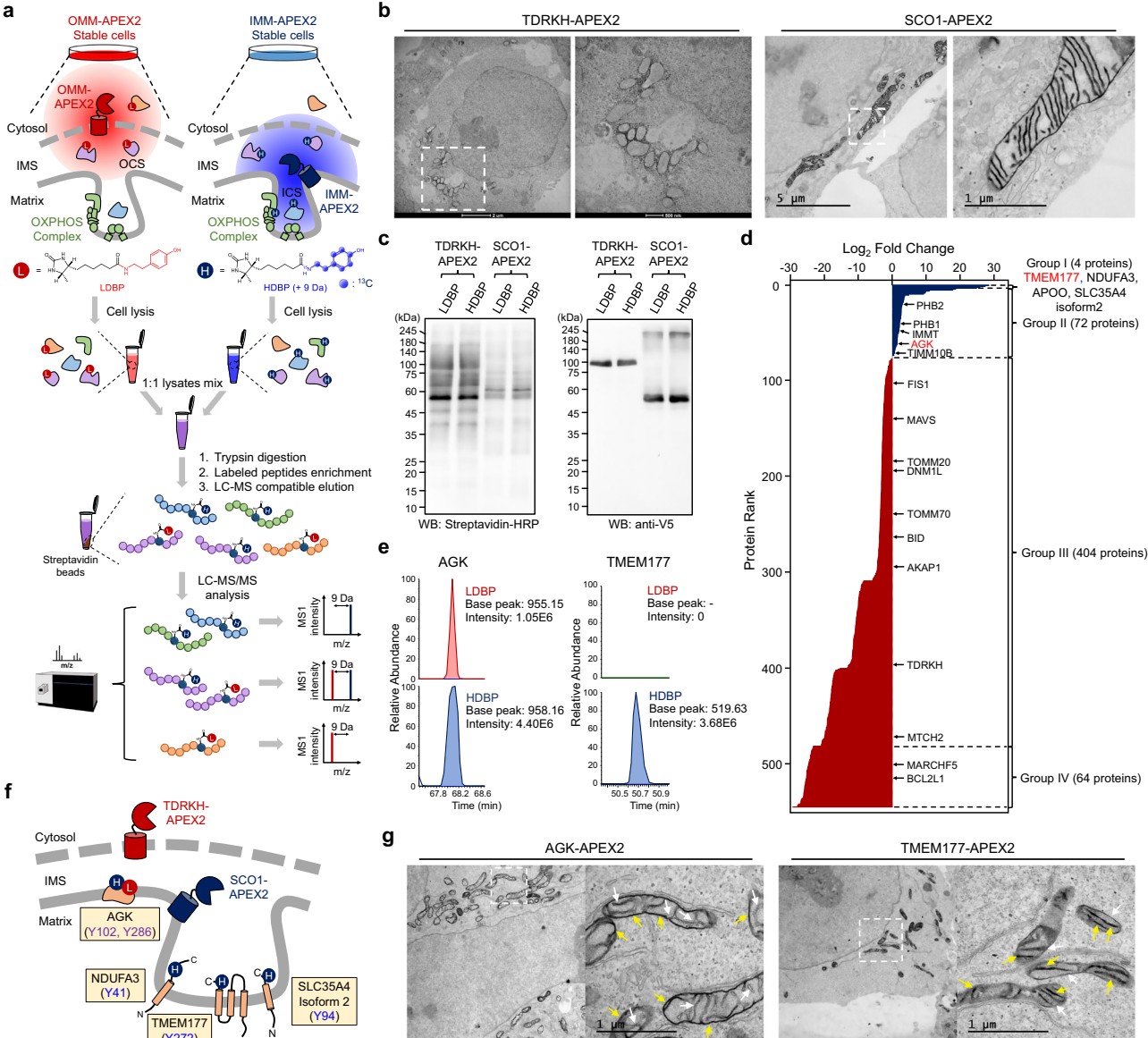

**Fig. 2 | Duplexed super-resolution proximity labeling analysis using isotope-coded phenol probes for APEX labeling and its application in ICS proteome profiling. a** Strategy for intracristal space (ICS) proteome mapping using desthiobiotin-conjugated isotope-coded phenol probes for APEX labeling (ICAX). Light desthiobiotin-phenol (LDBP) modified proteins labeled with outer mitochondrial membrane (OMM)-APEX2 were mixed with heavy desthiobiotin-phenol (HDBP) modified proteins labeled with inner mitochondrial membrane (IMM)-APEX2 after cell lysis. The mixed proteome was digested and subjected to the enrichment procedure of labeled peptides using streptavidin beads, followed by liquid chromatography-mass spectrometry (LC-MS)/MS analysis for isotopic MS1 quantification. **b** Electron microscopic (EM) images of TDRKH-APEX2 (OMM-APEX2) and SCO1-APEX2 (IMM-APEX2) after DAB staining. The boxed region is expanded in the image on the right. **c** Labeled proteins were analyzed via western

blotting with streptavidin-HRP. The levels of TDRKH-APEX2 and SCO1-APEX2 were confirmed via western blotting with an anti-V5 antibody. **d** Bar graph showing the FC values of proteins labeled with TDRKH-APEX2 (LDBP) and SCO1-APEX2 (HDBP); groups I–IV (a total of 544 proteins). OMM and IMM marker proteins are indicated according to MitoCarta 3.0. *n* = 3 biological replicates. **e** MS1 intensity of AGK and TMEM177, labeled with LDBP (TDRKH-APEX2) and HDBP (SCO1-APEX2), respectively. **f** Membrane topology and labeled sites of the representative ICS and outer ICS (OCS) proteins. **g** EM images of AGK-APEX2 and TMEM177-APEX2 after diaminobenzidine (DAB) staining. The white dashed box indicates the region of interest, which is enlarged in the adjacent image. White arrows indicate the mitochondrial membrane stained with $OsO_4$, and yellow arrows indicate the DAB precipitate stained with $OsO_4$.

APEX2 construct, APEX2 was conjugated to the C-terminus of SCO1 facing to the IMS (Fig. 2b)[18]. Electron microscopy (EM) images confirmed that SCO1-APEX2 was not restricted to any specific IMM subdomain, allowing coverage of both the ICS- and OCS-localized proteomes. For the OMM-APEX2 construct, we used TDRKH, a protein known to localize on the OMM, and its C-terminus faces to the cytosol (Fig. 2b)[30]. EM images of the TDRKH (1–50 aa)-APEX2 construct confirmed the strong OMM-targeting specificity of the TDRKH N-terminal transmembrane sequence, which included a predicted transmembrane domain (Supplementary Fig. 5a–d). TDRKH is not included in the TOM translocase complex that minimizes the risk of mislabeling matrix- or IMS-targeting proteins by APEX2 during protein import[31].

We confirmed that LDBP and HDBP exhibited the same reactivity toward SCO1-APEX2 (Fig. 2c); no differences were observed between their probe labeling patterns (Supplementary Fig. 5e). TDRKH-APEX2 also exhibited similar reactivity between HDBP and LDBP. Using these probes, we conducted ICAX-labeled site identification using stable cell lines expressing TDRKH-APEX2 and SCO1-APEX2, respectively. In triplicate biological experiments, we identified 544 labeled proteins originating from DBP-labeled peptides, which exhibited high reproducibility among replicates, and confidently obtained the log2 fold change (FC) between HDBP and LDBP (Supplementary Data 6). These results support the reliability of the ICAX method for proteome quantification.

As the FC of each labeled protein indicated its differential accessibility to phenoxyl radicals from SCO1-APEX2 and TDRKH-APEX2, we classified the ICAX-labeled proteins into four sub-mitochondrial populations (groups I–IV) based on their FC_SCO1/TDRKH values (Fig. 2d). In group I (log2 FC_SCO1/TDRKH > 20), only four proteins (TMEM177, NDUFA3, APOO, and SLC35A4 isoform 2) were exclusively labeled with IMM-APEX2 and considered strong ICS protein candidates (Fig. 2e, f). Among them, TMEM177-APEX2 displayed an exclusive, ICS-localized, DAB staining pattern in EM images (Fig. 2g and Supplementary Fig. 6a, b). These results indicate that TMEM177 is an ICS-localized protein and that its localization is not altered by tagging other proteins, including APEX2.

Given the intriguing localization of TMEM177, we performed further experiments to determine its function in ICS. First, we generated TMEM177-knockdown (KD) cells using a green fluorescent protein (GFP)-expressing shRNA viral vector (Supplementary Fig. 7a). We then performed correlative light and electron microscopy to visualize mitochondrial morphology. TMEM177 KD cells displayed loss of mitochondrial cristae density when compared with non-infected cells (Supplementary Fig. 7b). Basal and maximal (i.e., FCCP-treated) oxygen consumption rate (OCR) levels were significantly lower in TMEM177 KD cells than in shControl cells (Supplementary Fig. 7c). These findings indicate that TMEM177 is highly involved in the formation of cristae and functional respiration of mitochondria[32].

Notably, group-II proteins (0 < log2 FC_SCO1/TDRKH < 20, 72 proteins) include proteins localized in the outer ICS (OCS), i.e., the space between the OMM and the inner boundary membrane, and their sub-IMS localization cannot be systematically accessed using conventional approaches. We further examined AGK localization using APEX2 conjugation. In EM imaging experiments, we observed clear OCS localization of AGK-APEX2 (Fig. 2g and Supplementary Fig. 8a, b). The OCS localization of AGK may correlate well with its known function in protein transport between the IMM and OMM as a subunit of the TIM22 complex[33]. Additionally, we revealed the membrane topology of the OMM based on the FC values of group-II and III proteins (Supplementary Fig. 8c, d). For example, the FC_SCO1/TDRKH of the labeled sites on OMM proteins, including VDAC1-3 and SAMM50, clearly showed that these beta-barrel proteins should have a single determined membrane topology at the OMM in live cells (Supplementary Fig. 8c, d), which contradicts the previous observation regarding the random orientation of VDAC in in-vitro measurements[34,35]. Overall, our log2 FC values of ICAX-labeled proteins represent the vertical ordering of mitochondrial protein complexes in the order of ICS, OCS, and OMM.

## ICS and OCS proteome mapping using ICAX

Since SCO1 is not an ICS-specific protein, we aimed to reveal more ICS and OCS proteomes using ICAX on TMEM177-APEX2 and AGK-APEX2 stable cell lines (Fig. 3a, b). Consequently, we obtained 304 LDBP and HDBP-modified proteins that exhibited reproducible FC_TMEM177/AGK values among replicates. FC_SCO1/TDRKH and FC_TMEM177/AGK were broadly correlated (Fig. 3c). We found that cytosolic (non-mitochondrial), OMM-, and mitochondrial-associated membrane-localized proteins were labeled more strongly by AGK-APEX2, as AGK is located closer to the porous OMM and cytosolic compartments, than by TMEM177. By contrast, proteins labeled preferentially by TMEM177-APEX2 included subunits of OXPHOS complexes, and their assembly factors, which indicated that these are candidate ICS proteins (Supplementary Fig. 9a–d).

We generated an ICS/OCS map based on FC_TMEM177/AGK and we selected 64 proteins that were highly enriched by TMEM177-APEX2 (group ICS) according to the FC values of OPA1 and YME1L1, as these proteins localize at the cristae junction for regulation of cristae structures (Fig. 3c). Among the ICS group proteins, we confirmed the ICS-specific localization of EndoG and TMEM126B using confocal and APEX-EM imaging (Fig. 3d, e). Additionally, several IMS-side proteases (e.g., OMA1, CLPB, HTRA2, PARL, and YME1L1) and MCU complex subunits (e.g., MICU1 and MICU2) were highly labeled with TMEM177-APEX2 (Supplementary Fig. 9c), indicating that they play a role in mitochondrial quality control and calcium transport in the ICS, respectively.

Our analysis of the TMEM177/AGK-APEX2 samples using ICAX revealed the presence of both the ICS and OCS proteome. The OCS group (58 proteins), enriched with peptides intensively labeled with AGK-APEX2, contained numerous metabolite transporter proteins (SLC25 family proteins), ADP/ATP translocase (ANP) and conversion enzymes (e.g., RDH13, AK2, GPD2, and PTGES2). These findings suggest that the OCS may play a role in controlling metabolic flux at the OMM–IMM interface[36]. A detailed list of the identified OCS and ICS proteins with membrane topological information[37–40] is provided in Supplementary Figs. 10 and 11 and Supplementary Data 7.

Since we identified TMEM177 and AGK as representative marker proteins in the ICS and OCS, respectively, we further used them to target fluorescent protein (FP) sensors, such as pHluorin2 (pH)[41], HyPer7 (redox)[42], and gTEMP[43] (temperature), to the ICS and OCS to selectively assess the microenvironment (Supplementary Fig. 12a). We measured the pH values of the ICS (pH = $6.9 \pm 0.3$, mean ± SD) and OCS (pH = $7.2 \pm 0.2$, mean ± SD), which were lower than those in the mitochondrial matrix (pH = $8.2 \pm 0.1$, mean ± SD) and cytosol (pH = $7.4 \pm 0.4$, mean ± SD), using pHluorin2[41] (Supplementary Fig. 12b, c) and found them to be consistent with previous findings showing that the ICS has an acidic environment compared with that of the other mitochondrial compartments[44].

We measured hydrogen peroxide level in the ICS and OCS by fusing HyPer7[42] with TMEM177 and AGK, respectively, and found that the OCS had a comparatively oxidized environment compared to the ICS (Supplementary Fig. 12d). This result corroborates with our finding on CHCHD4 in OCS, which is known to form multiple disulfide bonds under physiological conditions involved in oxidative folding system[45]. We also detected a prompt response of TMEM177-conjugated gTEMP[43], a thermal sensing FP, under uncoupler treatment (i.e., BAM15), suggesting that the heat generated from mitochondria[46] can accumulate in the ICS during the uncoupling process (Supplementary Fig. 12e).

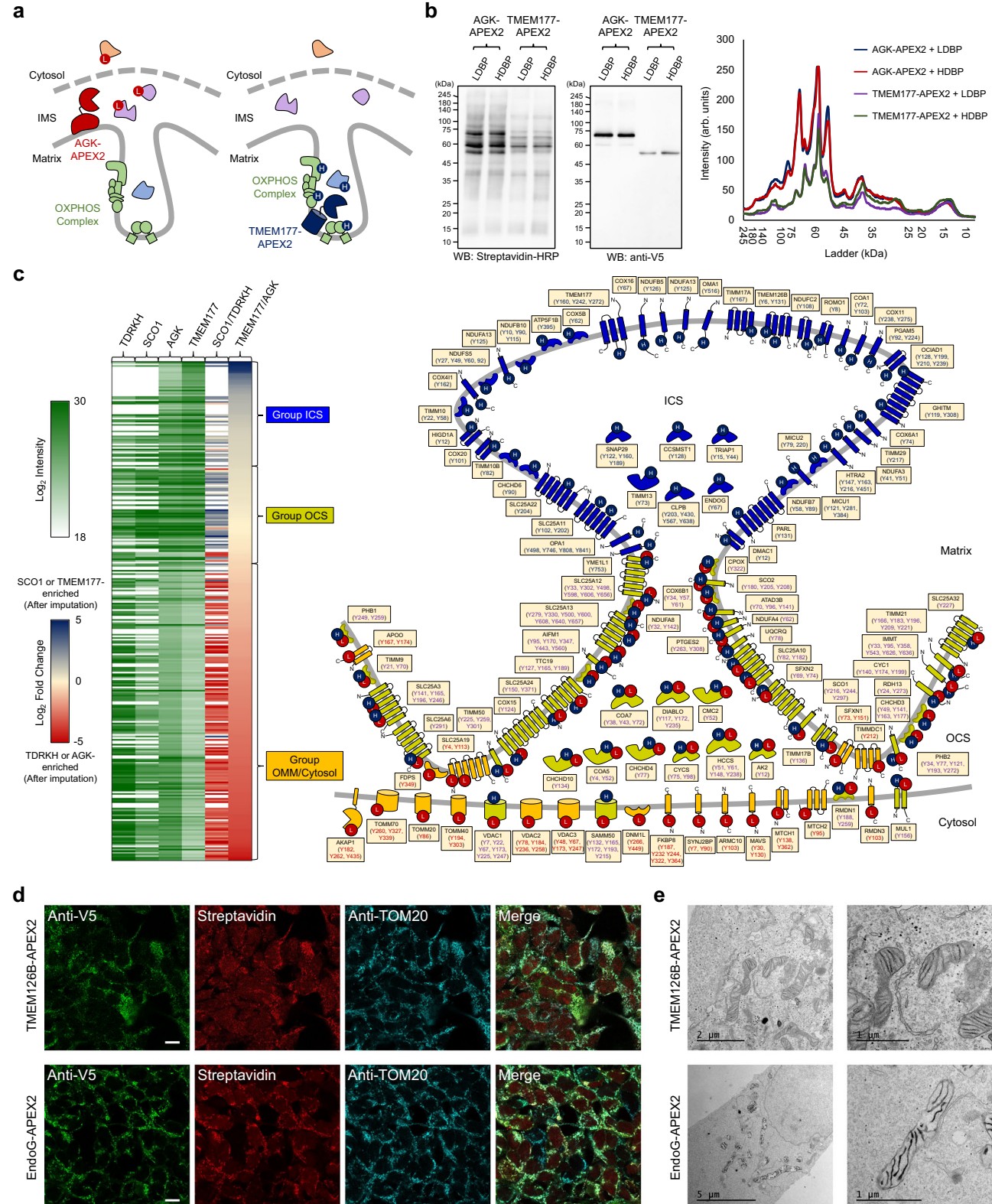

**Fig. 3 | ICS proteome mapping using AGK- and TMEM177-APEX2. a** Schematic representation showing TMEM177-APEX2 and AGK-APEX2 labeling using isotope-coded phenol probes for APEX labeling (ICAX) for the identification of intracristal space (ICS) and outer ICS (OCS) proteins, respectively. **b** Streptavidin-HRP western blot pattern of light desthiobiotin-phenol (LDBP)- and heavy desthiobiotin-phenol (HDBP) proteins labeled with AGK-APEX2 and TMEM177-APEX2. APEX2 expression levels were detected using an anti-V5 antibody. **c** Heat map-based comparison of the TMEM177/AGK (n = 4 biological replicates) and SCO1/TDRKH datasets. The ICS,

OCS, and outer mitochondrial membrane (OMM)/Cytosol groups were classified based on fold-change values; the inner mitochondrial membrane, intermembrane space, and OMM proteins in each group are illustrated according to their known topology in the right panel, with modification sites from the TMEM177/AGK dataset. A blank space indicates that no proteins were detected in the primary SCO1/TDRKH dataset. **d** Confocal images of TMEM126B-APEX2 and EndoG-APEX2 obtained after immunostaining. Scale bars represent 10 μm. **e** Localization of TMEM126B and EndoG validated through APEX electron microscopy imaging.

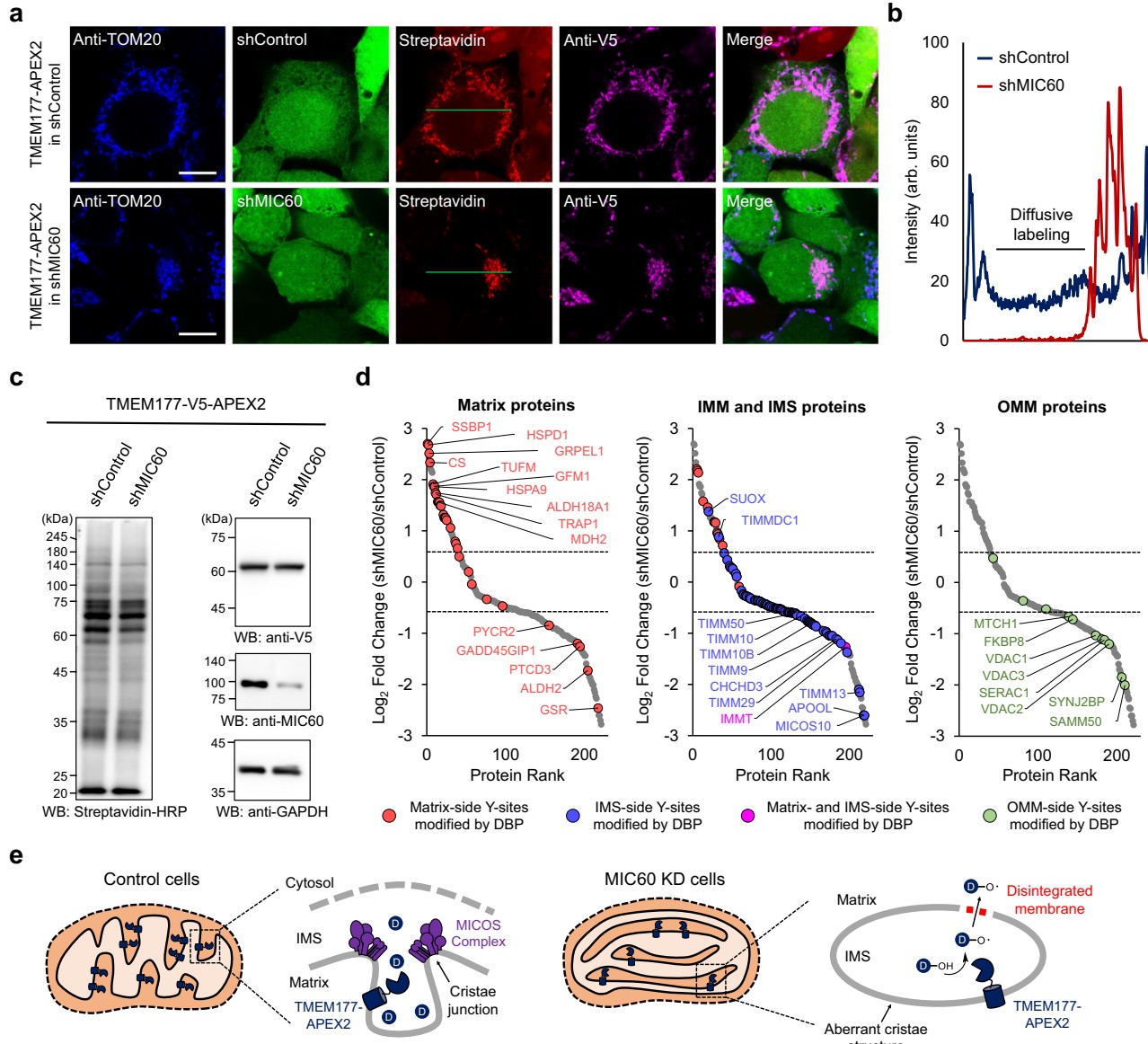

**Fig. 4 | Proteome mapping in the aberrant ICS induced by the MICOS complex inhibition. a** Confocal imaging results showed aggregated mitochondria in the MIC60 KD cells. Anti-V5 and anti-TOM20 antibodies were used to visualize TMEM177-APEX2 and mitochondria, respectively. Scale bars represent 10 μm. **b** Line scan analysis for region of interest (green line) in (**a**) shows diffusivity of DBP radicals. (**c**) Biotinylation level in MIC60 knockdown (KD) cells was detected by western blotting using streptavidin-HRP. TMEM177-V5-APEX2 expression level was visualized using an anti-V5 antibody, and GAPDH was used as a loading control.

Reduced expression level of MIC60 in KD cells were determined by anti-MIC60 antibody. **d** Scatter plot showing the log2-fold change of the identified proteins in shMIC60 compared with shControl cells (*n* = 2 biological replicates). Identified proteins were highlighted according to their localization and the modification sites in the mitochondria. **e** Proposed model demonstrating that the inhibition of mitochondrial contact site and cristae organizing system (MICOS) complex leads to disintegration of membrane structures in aberrant intracristal space.

## Identification of spatial proteome in aberrant cristae induced by mitochondrial contact site and cristae organizing system (MICOS) complex inhibition

The mitochondrial structure in MIC60 (also known as IMMT)-deficient cells has been well characterized by EM imaging. Recently, various studies demonstrated that mitochondria in MIC60-deficient cells exhibit a very low frequency of cristae junctions, leading to aberrant cristae structures[47]. Based on these findings, we generated control and MIC60 knockdown (KD) cell lines expressing TMEM177-APEX2 using the shRNA to examine the proteome surrounded by aberrant cristae caused by inhibition of the MICOS complex.

We validated the biotinylation pattern of TMEM177-APEX2 in shControl and shMIC60 cells using western blotting and confocal imaging. We observed mitochondrial aggregation in MIC60 KD cells

consistent with previous reports in MIC60 KO mice[48]. The TMEM177-APEX2 biotinylation pattern in MIC60 KD cells was comparatively restricted (Fig. 4a, b), suggesting that DBP radicals were confined within the aberrant ICS. Additionally, several biotinylation bands in MIC60 KD cells were slightly weaker than those corresponding to the control cells (Fig. 4c), which may be owed to the restricted diffusion of the labeling.

Then, we conducted ICAX analysis with control and MIC60 KD cells expressing TMEM177-APEX2 and identified 221 proteins, including 33 matrix proteins, 91 IMM proteins, and 15 IMS proteins[49,50]. We found that subunits of the TIM and MICOS complexes, labeled in the IMS, and OMM proteins were enriched in the control cells (Fig. 4d). We also matched with our OCS dataset, consequently confirming that OCS proteins were enriched in the control cells (Supplementary Fig. 13a).

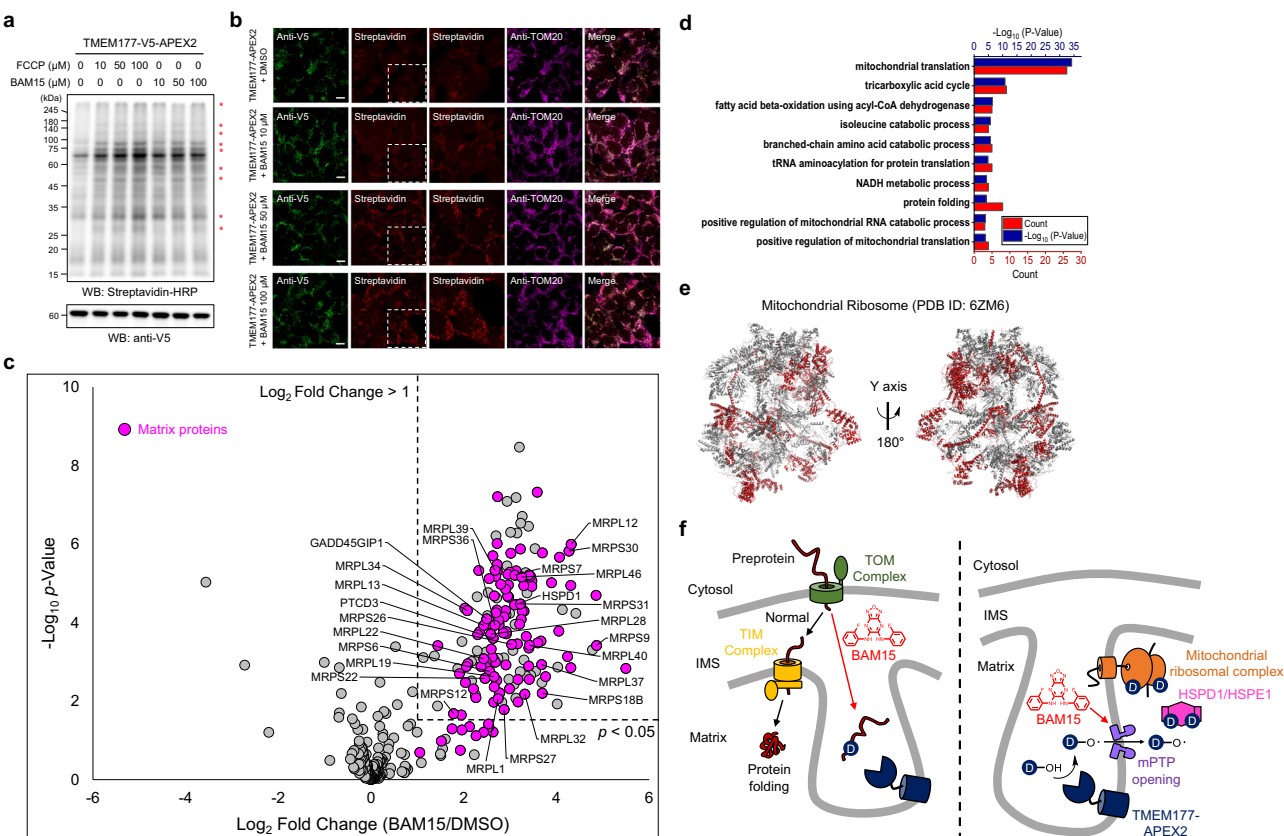

**Fig. 5 | Mitochondrial dynamics under the mitochondrial uncoupling process.**
**a** Western blot analysis of proteins labeled with desthiobiotin-phenol (DBP) using
TMEM177-APEX2 after uncoupler treatment (e.g., FCCP or BAM15). Additional
biotinylated bands are indicated by asterisks. TMEM177-APEX2 levels were analyzed
using an anti-V5 antibody. **b** Confocal images showing the expression patterns of
TMEM177-APEX2 and those of its biotinylated proteins with or without BAM15
treatment. Biotinylated proteins were stained with streptavidin-conjugated AF568,
and APEX2 was detected using an anti-V5 antibody. Fluorescence intensity is indi-
cative of the biotinylation signal intensity. Scale bars represent 10 μm. **c** Volcano
plot analysis of proteins labeled with ICAX probes using TMEM177-APEX2 under
BAM15 or DMSO (control) treatment ($n = 4$ biological replicates). The $p$ value was
obtained using an unpaired two-tailed Student's $t$-test. Mitochondrial matrix

proteins are displayed in pink dots, and mitochondrial ribosome subunits and
mitochondrial heat shock proteins (HSPD1 and HSPE1) are indicated by gene
names. **d** DAVID functional Gene ontology analysis[54] of biological processes for the
detected mitochondrial matrix proteins shown in (**c**). The $p$ value was obtained
using a Modified Fisher's Exact Test. The red bar indicates the number of proteins
involved in the described processes, and the dark blue bar indicates the $-\log_{10} p$
value. **e** Mitochondrial ribosomal subunits (24/111, red color) that were strongly
labeled with TMEM177-APEX2 under uncoupler treatment are shown in the cryo-
electron microscopy structure of the mitochondrial ribosome (PDB: 6ZM6)[55].
Nucleic acid chains and ligands have been omitted for clarity. **f** Proposed possible
models of the mitochondrial intracristal subdomain dynamics during the mito-
chondrial uncoupling process.

These results indicate that DBP radicals generated by TMEM177-APEX2
were inaccessible in the OCS and cytosol owing to the formation of an
aberrant ICS.

The biogenesis of the OXPHOS complexes is highly dependent on
TIM23 complex, which directly inserts most of the subunits laterally[51].
However, this process can be disrupted in aberrant ICS by a dis-
connection with the TIM23 complex at the inner boundary membrane.
In this context, we found that several subunits of the OXPHOS com-
plexes showed reduced labeling intensity in MIC60 KD cells (Supple-
mentary Fig. 13a), which corresponds to the attenuated OCR observed
in MIC60-deficient cells[47].

Conversely, 23 mitochondrial matrix proteins and 12 IMM pro-
teins with matrix-side DBP modifications (Supplementary Fig. 13b)
exhibited strong labeling with TMEM177-APEX2 in MIC60 KD cells
(Fig. 4d). These results suggest that IMM integrity was compromised,
allowing increased permeability to DBP radicals resulting from the
disordered cristae structure (Fig. 4e).

**Mitochondrial uncouplers remodel intracristal space proteome
and membrane permeability**
Next, we determined whether ICAX can sensitively detect dynamic
changes in spatial proteomic components or the spatiome[8] at a single

location. For this experiment, we used the TMEM177-APEX2 stable cell
line to monitor the changes in ICS proteome under conditions of
energy stress. For inducing stress conditions, we treated cells with
uncouplers, such as FCCP and BAM15, which alter the IMM
potential[46,52,53] and enhance the exothermic mitochondrial oxygen
consumption rate, which can increase the local mitochondrial
temperature[46]. Indeed, additional proteins were labeled with
TMEM177-APEX2 when the cells were pre-incubated with BAM15 or
FCCP for 1 h (Fig. 5a, b). We used an MTS-APEX2 proximal biotinylation
assay and EM imaging to confirm that TMEM177 membrane topology
did not change upon BAM15 treatment (Supplementary Fig. 14a, b),
which suggested that uncouplers affect the proximal proteome net-
work of TMEM177-APEX2 in the ICS.

These additionally labeled proteins were analyzed using ICAX.
TMEM177-APEX2 stable cells were co-treated with BAM15 and HDBP for
1 h, followed by APEX labeling. As a control, LDBP was used with the
DMSO-treated sample. LC-MS/MS analysis of BAM15 treated/untreated
samples of ICAX-labeled peptides revealed 111 mitochondrial matrix
proteins (log2 FC_BAM15/DMSO > 1, $p < 0.05$) that were significantly
labeled with TMEM177-APEX2 after BAM15 treatment (Fig. 5c, Supple-
mentary Data 9). Among these proteins, mitochondrial ribosomal
subunits[54,55] (24/111, 21.6%) were highly enriched with TMEM177-APEX2,

compared with other abundant mitochondrial matrix-targeting proteins, during the uncoupling process (Fig. 5d, e).

We also found that HSPD1, a mitochondrial matrix-targeting chaperone, was strongly enriched by TMEM177-APEX2 following BAM15 treatment in LC-MS/MS analysis (Fig. 5c). Western blot analysis confirmed that HSPD1 labeling intensity significantly increased after BAM15 treatment in TMEM177-APEX2 samples (Supplementary Fig. 14c). We also conducted a mitochondria isolation experiment to determine the localization of proteins labeled by TMEM177-APEX2, as labeled mitochondrial proteins accumulated in the cytosol during the mitochondrial uncoupling process[53,56]. As a result, we found that most of the biotinylated proteins by TMEM177-APEX2, including HSPD1, were present in the mitochondria, not in the cytosol (Supplementary Fig. 15a).

Next, we used puromycin and cycloheximide (CHX) to inhibit the accumulation of proteins in the cytosol, including TMEM177-APEX2 and mitochondrial matrix proteins. To monitor protein synthesis inhibition, we pre-treated cells with 35 μM of puromycin and CHX for 2 h and added azidohomoalanine (AHA) for 30 min. We found that protein synthesis was efficiently blocked by puromycin and CHX (Supplementary Fig. 15b). Next, we pre-treated cells with 35 μM of puromycin and CHX for 2 h and added BAM15 and LDBP for labeling to determine whether HSPD1 is labeled by TMEM177-APEX2 in the mitochondria. Then, we found that puromycin and CHX treatment did not change the HSPD1 biotinylation level (Supplementary Fig. 15c). These results indicate that biotinylation of mitochondrial matrix proteins by TMEM177-APEX2 occurs in the mitochondria under the uncoupling process.

The mitochondrial matrix proteins were not significantly labeled with OCS-localized AGK-APEX2 (Supplementary Fig. 16a, b), which suggested the existence of an ICS-specific event. Recent studies have revealed that mitochondrial matrix proteins, including ribosomal subunits, are accumulated in the IMS of compromised mitochondria upon inhibition of ATP synthase in *yeast*, potentially preventing their mistargeting to the nucleus[57,58]. Therefore, the high levels of mitochondrial ribosomal subunits and HSPD1 detected using TMEM177-APEX2 can be interpreted as the temporal accumulation of mitochondrial matrix proteins in the ICS under uncoupled conditions (Fig. 5f). As several IMS-localized mitochondrial proteases (e.g., PARL and YME1L1) are also found in the ICS, the ICS may be an active space for mitochondrial proteostasis under uncoupling or thermogenic conditions[46], which requires further validation.

Another possibility is that the mitochondrial permeability transition pore (mPTP) opening was induced by a protonophore in the ICS[59–62]. The phenoxyl radical probe could pass the membrane through mPTP from the ICS to the mitochondrial matrix and modify the matrix proteins (Fig. 5f). To examine whether DBP modification occurs in the mitochondrial matrix, we employed MTS-dsRed, which undergoes MTS cleavage by mitochondrial processing peptidase (MPP) in the matrix. Consequently, we found that MPP-processed dsRed was strongly labeled by TMEM177-APEX2 following BAM15 treatment (Supplementary Fig. 16c), suggesting that DBP radical cross the IMM. Each model has the potential to enhance our understanding of mitochondrial ICS dynamics, although further studies are warranted to test these hypotheses.

## Discussion

In this study, we introduce ICAX, an isotope-coded affinity enrichment approach that enables sensitive, reproducible, and quantitative proximity proteomics at sub-organelle resolution. Using isotope-coded probes in combination with APEX2 targeted to the IMS, we achieved high coverage profiling of two distinct IMS sub-compartments, defined as ICS and OCS.

We demonstrated the applicability of ICAX for dynamic proteome analysis under physiological perturbations induced by MICOS

complex inhibition and mitochondrial uncoupling, respectively. In both conditions, we observed that a significant number of mitochondrial matrix proteins were labeled by TMEM177-APEX2, possibly due to DBP radical diffusion across the IMM from the ICS. This observation raises interesting questions regarding the regulation of membrane permeability within the ICS under various stress conditions.

The ICAX method exhibited superior performance in peptide coverage compared to conventional isobaric labeling strategy using Tandem mass tag (TMT) labeling on DBP-modified peptides (Supplementary Fig. 17). TMT labeling workflows typically require additional sample handling steps, which can introduce technical variability and sample loss, particularly in the context of low-abundance biotinylated peptides (Supplementary Fig. 17a, b). In contrast, the original ICAX protocol employs a streamlined peptide-level enrichment strategy combined with LC-MS, which is optimized to minimize the sample manipulation[11]. As a result, the ICAX approach preserved more peptides, leading to the identification of a greater number of biotinylated proteins by MTS-APEX2 (321 proteins) compared to the TMT-labeled counterpart with same construct (243 proteins; Supplementary Fig. 17c, d). Despite the aforementioned issues, combining ICAX probes (LDBP and HDBP) with TMT remains a valuable approach, as it effectively expands the multiplexing capability for multiple sample analysis.

It is noteworthy that our data conflicts with the current understanding of sub-mitochondrial localization of certain proteins. Although the TIM complex is generally considered to reside in the OCS to facilitate mitochondrial protein import, we observed differential localization of its components such as TIM9 in the OCS and TIM10 positioned in the ICS. Given that TIM9 and TIM10 typically form a hetero-hexamer in which surface-exposed tyrosine residues are mutually masked, we believe that our data suggests the presence of monomeric or unassembled TIM10 pool in the ICS. Unassembled TIM10 are known to be degraded by Yme1[63], whose human homolog (YME1L1) is an ICS protein in current study. Further study requires to identify ICS-targeting mechanism of these proteins. Similarly, we observed that complex III subunits were identified in the OCS, whereas subunits of OXPHOS complexes I, IV, and V were predominantly enriched in the ICS (Supplementary Fig. 9d). Since ICS proteins must transit through the OCS during the import process, it is possible that certain complex III subunits are transiently retained in the OCS. During this retention, their tyrosine residues may become more surface-exposed compared to assembled complex, facilitating their modification by AGK-APEX2.

To address this issue, complementary PL methods capable of labeling different amino acid residues should be employed to validate our findings on ICS and OCS proteomes. These collective data may provide deeper insights into the spatial distribution of the mitochondrial proteome. We believe that our proof-of-concept for ICAX is broadly adaptable in other proximity labeling tools. For instance, LDBP and HDBP are potentially applicable to photocatalyst-mediated proximity labeling that generates phenoxyl radicals[64]. Furthermore, the ICAX framework is compatible with other proximity labeling platforms that employ diffusible reactive intermediates such as quinone methide[42,43], tyrosinase[44], singlet oxygen[65], and carbene[6,7], offering a generalizable strategy for quantitative proximity proteomics in diverse biological contexts.

## Methods
### Plasmids and cloning
Genes were cloned into the specified vectors using standard enzymatic restriction digest and ligation with T4 DNA ligase. PCR products were digested by restriction enzymes and ligated into cut vectors. In all cases, the CMV promoter was used for expression in mammalian cells. The genetic constructs cloned and used in this study are summarized in Supplementary Data 11. In this table, Protein processed size during

translocation was obtained by programs: Mitoprot[66] (http://ihg.gsf.de/ihg/mitoprot.html or Mitoprot software is available by: ftp://ftp.biologie.ens.fr/pub/molbio).

## Stable cell line selection and culturing

Flp-In T-Rex 293 cells (Thermofisher Scientific, R78007) were cultured in DMEM (Gibco) supplemented with 10% FBS, 2 mM l-glutamine, 50 units/ml penicillin, and 50 μg/ml streptomycin at 37 °C under 5% $CO_2$. Cells were grown in a T25 flask. Stable cell lines were generated by co-transfection with the pcDNA™5/FRT/TO expression construct and the pOG44 plasmid. Cells were transfected at 60–80% confluence using Lipofectamine 2000 (Thermofisher Scientific, 11668019) or polyethylenimine (Polysciences, 23966-1), typically with 20 μL Lipofectamine 2000 and 4000 ng plasmid (9:1 = pOG44:pcDNA5) per T25 flask. The growth medium was refreshed after 4 h from transfection. After 24 h, cells were split into a 90 mm cell culture dish (SPL, 11090) with the proper concentration of hygromycin B (100–200 μg/ml, Thermofisher Scientific, 10687-010). Media containing hygromycin B were changed every 3–4 days. After 2–3 weeks, 3–4 colonies were selected and transferred to a 24-well plate. Cells were continuously split into larger plates, and cell stocks were made. After splitting the cells into a 6-well plate, separate samples were prepared for expression testing. To induce the protein expression in stable cells, cells at 60-80% confluence were treated with 100 ng/ml doxycycline (Sigma Aldrich).

## Lentiviral infection for generating knockdown cell lines

HEK293T cells (ATCC) were cultured with DMEM in a 6-well plate. Cells were transfected at 60–80% confluence using Lipofectamine 2000 or polyethylenimine. A mixture of pGIPZ plasmid (1000 ng) as transfer vector, psPAX2 (750 ng) as packaging vector, and pMD2.G (250 ng) as envelope vector was used for lentivirus generation. The growth medium was refreshed after 4 h from transfection. After 24 h from transfection, media was collected and filtered by a 0.22 μm PES syringe filter (JetBiofil). Filtered media was added into Flp-In T-Rex 293 cells or non-transfected HEK293T cells for infection. The selection was performed by adding puromycin (1 μg/ml) for 24 h.

## Immunofluorescence for confocal imaging

Stable cell lines (or other culture cell lines) were cultured on an 18 mm coverslip (Marienfeld) in a 12-well plate. Cells at 50-70% confluence were treated with 100 ng/ml doxycycline for protein expression. For APEX2 stable cell lines, cells were treated with 250 μM LDBP or HDBP containing fresh cell growth media after 18-24 h from doxycycline treatment. Then, APEX2 labeling was performed by the addition of $H_2O_2$ (Sigma Aldrich H1009) at a final concentration of 1 mM for 1 min. The reaction was quickly quenched by adding 2x quenching solution, which contains 10 mM Trolox, 20 mM sodium azide, and 20 mM sodium ascorbate in Dulbecco's Buffered Saline (DBPS). Cells were washed with 1x quenching solution twice more and fixed with paraformaldehyde for 10 min. Cells were permeabilized with cold methanol for 5 min at −20 °C after washing with DPBS twice. Cells were again washed with DPBS to remove residual methanol. Blocking was performed by using a 2% BSA in 1x Tris-Buffered Saline containing 0.1% Tween 20 (TBST). Then, cells were incubated with the primary antibody in a blocking solution for 1 h, followed by washing with 1x TBST three times each for 5 min. Cells were incubated with a secondary antibody in the blocking solution (1:3000 dilution, 0.33 μg/ml) for 30 min. Streptavidin-AF568 or streptavidin-AF647 were treated along with secondary antibodies. After washing with 1x TBST, images were acquired using a confocal laser scanning microscope (SP8 X, Leica) with an objective lens (HC PL APO ×100/1.40 oil), white light laser (470–670 nm, 1 nm tunable laser), and a HyD detector. The system was operated using LAS X software. Line scan analysis was performed using Fiji software (Image J, NIH)[67].

## Western blotting

Stable cell lines were cultured with DMEM in a 12-well or 6-well plate. Cells at 60-80% confluence were treated with 100 ng/ml doxycycline for 18-24 h. After the APEX2 reaction, cells were lysed with RIPA lysis buffer (ELPIS-BIOTEECH) containing a protease inhibitor cocktail on a microtube rotator. 100 μl and 200 μl lysis buffer were used for the 12-well and 6-well plate, respectively. The concentration of extracted proteins was measured by BCA assay, and the protein concentration of each sample was adjusted equally by using RIPA lysis buffer. Then, proteins were denatured by heating for 5 min after adding an SDS loading buffer. Proteins were loaded onto 10% SDS-PAGE gel and ran at 200 V for 1 h. Next, proteins were transferred to the nitrocellulose membrane (Pall Corporation) at 25 V constant voltage for 6 h. Blocking was performed with 2% skim milk (in 1x TBST) for 1 h, and the membrane was incubated with a primary antibody in 2% skim milk overnight. Membrane washing was conducted by 1x TBST four times each for 5 min. Then, the membrane was incubated with anti-mouse-HRP or anti-rabbit-HRP in 2% skim milk (1:3000 dilution, 0.33 μg/ml) for 1 h. For detecting biotinylated proteins, the membrane was incubated with streptavidin-HRP (ThermoFisher) in 1x TBST (0.2 μg/ml) after blocking. After washing the membrane four times with 1x TBST, the membrane was incubated with Clarity Western ECL Substrate solution (BioRad) for protein detection.

## Oxygen consumption rate (OCR) measurement

To measure OCR in TMEM177 knock-down cells, Seahorse XF cell mito stress test kit (Agilent, 103015-100) was used. Seahorse XF96 cell culture microplate (Agilent) was treated with 0.01% poly L-lysine overnight for coating. Poly L-lysine was removed, and the plate was washed by DPBS three times. Then, cells were cultured with DMEM for 24 h after cell counting. Cell growth medium was replaced with assay medium (Seahorse XF DMEM supplemented with 1 mM pyruvate, 2 mM glutamine, and 10 mM glucose) and incubated in a 37 °C non-CO2 incubator for 45 min. Compound solutions (each containing 1.5 μM oligomycin, 0.5 μM FCCP, and 0.5 μM rotenone/antimycin A, respectively) were loaded into the port of the pre-hydrated sensor cartridge. Sensor cartridge and microplate were placed on the instrument tray according to the manufacturer's protocol, and then OCR measurement was performed. Results were exported to Excel by using Seahorse XF Wave software v2.6 (Agilent).

## Mitochondrial isolation

Control cells and knockdown cells were grown in T175 flasks. Cells at 90% confluence were detached in DBPS using a cell scraper, and a cell pellet was prepared by centrifugation at 600 g at 4 °C for 5 min. The supernatant was removed, and 5 ml of cold IBc buffer (10 mM Tris–MOPS, 1 mM EGTA, and 200 mM sucrose) was added[68]. Cells were homogenized using a Dounce grinder (Sigma Aldrich) and stroked 20 times. The homogenate was transferred into a 15 ml polypropylene Falcon tube and centrifuged at 600 g for 10 min at 4 °C. The supernatant was collected into a new 15 ml polypropylene Falcon tube and centrifuged at 7000 g for 10 min at 4 °C. Supernatant was removed, and the pellet was washed with 500 μl of cold IBc. Mitochondrial pellets were prepared by centrifugation to homogenate again at 7000 g for 10 min at 4 °C.

## Immunoprecipitation of biotinylated proteins

Stable cell lines were cultured with DMEM in a 6-well plate and cells at 60-80% confluence were treated with 100 ng/ml doxycycline for 18-24 h. LDBP was treated along with BAM15 for 1 h, and APEX2 labeling was performed by addition of hydrogen peroxide at a final concentration of 1 mM for 1 min. Cells were lysed with RIPA lysis buffer containing proteases inhibitor cocktail. After concentration measurement by BCA assay, proteins were mixed with pre-washed streptavidin beads (Thermofisher Scientific, 88817) using a rotator at room

temperature for 1 h. 20 μl of streptavidin beads were used for 400 μg of extracted proteins. Then, streptavidin beads were isolated using a magnetic stand, and the supernatant (flow-through) was collected for western blotting. Non-specific binding proteins on the beads were washed using 0.1% SDS in 1x TBS buffer three times. The washing buffer was removed, and the SDS loading buffer was added for elution. Elution was performed by heating at 95 °C for 10 min.

## Measurement of ratiometric fluorescent sensors

Stable Flp-In T-Rex 293 cell lines expressing pHluorin2, HyPer7, and gTEMP were prepared for the measurement of intraceullar pH, hydrogen peroxide levels, and relative temperature, respectively. Each cell line was plated on 96-well black plates (SPL, Cat. No. 30296), and protein expression was induced by treatment with doxycycline. For pHluorin2 measurement, excitation scans (300 – 490 nm) with a fixed emission at 520 nm were performed using a Synergy ™ H1 microplate reader (BioTek Instruments, Inc.). The pH standard curve was generated using the cells incubated with the calibration buffers composed of 50 mM MES buffer (pH 6.0 and 6.5), 50 mM Tris–HCl (pH 7.0, 7.5, and 8.0). The pHluorin2 ratio was calculated by dividing the maximum values of fluorescence intensity at $380 \pm 10$ nm excitation by the intensity at $460 \pm 10$ nm excitation. For HyPer7, fluorescence intensity at 420 nm and 490 nm excitation with emission at 520 nm were measured. The HyPer7 ratio was obtained by dividing the fluorescence intensity at 490 nm excitation by the intensity at 420 nm excitation. For gTEMP measurement, emission scans with a fixed excitation at 360 nm were performed. The gTEMP ratio was calculated by dividing the fluorescence intensity at $510 \pm 4$ nm by the intensity at $424 \pm 4$ nm.

## DAB staining by APEX2 and transmission electron microscopy (TEM) imaging

Stable cell lines were cultured with DMEM in a 6-well plate, and cells at 60-80% confluence were treated with 100 ng/ml doxycycline to express APEX2 for 18-24 h. The cell growth medium was removed, and cells were incubated in a fixation solution (2% glutaraldehyde and 2% formaldehyde) at 4 °C for 1 h. All the process before detaching the cells was performed on ice or in a cold room. The fixation solution was quenched by washing with cold 20 mM glycine in 1x phosphate buffered saline (PBS) three times. Then, cells were treated with 3′–3′ diaminobenzidine (DAB, Sigma Aldrich, D8001) solution (2.3 mM DAB and 10 mM $H_2O_2$ in 1x PBS) for 10-20 min. Cells were washed with 1x PBS three times and incubated with 2% osmium tetroxide for 40 min. Cells were washed with cold TDW five times each for 2 min and detached by using a cell scraper. Cell pellets were prepared in a microtube by centrifugation at 1000 g at room temperature for 5 min. Cells were dehydrated by sequential addition of ethanol solutions (with this order: 70%, 80%, 90%, and 100% ethanol x 3) each for 20 min. EMbed 812 kit (Electron Microscopy Sciences) was used to prepare resin. Then, cells were infiltrated by sequential treatment of 1:1 ethanol and resin solution for 2 h, 1:2 ethanol and resin solution overnight, and 100% resin solution for 30 min. The solution was replaced with 500 μl fresh resin solution and incubated oven at 60 °C for 48 h. Embedded cells were cut into 50-100 nm sections using a diamond knife-equipped ultramicrotome. Sections were loaded on the grid, and double staining was performed with uranyl acetate and lead citrate. TEM imaging was taken with a bio-transmission electron microscope at Korea Basic Science Institute (KBSI).

## Correlative Light and Electron Microscopy (CLEM)

CLEM was performed by following procedure[47]. Stable cells were grown in 35 mm glass grid-bottomed culture dishes (MatTek, P35G-1.5-14-CGRD) to 30–40% confluency. Next day, live-cell images were acquired with a confocal light microscope (Ti-RCP, Nikon, Japan) at 37 °C in a temperature and CO2-controlled chamber. Cells were fixed with 2.5% glutaraldehyde (Electron Microscopy Sciences, 16210) and

2% paraformaldehyde (Electron Microscopy Sciences, 19210) in 0.1 M cacodylate solution (pH 7.0; Merck, C0250) for 1 h at 4 °C. After being washed, the cells were dehydrated with a graded ethanol series and infiltrated with an Embed-812 embedding kit (Electron Microscopy Sciences, 14120) and polymerized in an oven at 60 °C for 48 h. After embedding, 60 nm sections were cut horizontally to the plane of the block by ultramicrotome (Leica Microsystems, Germany, UC7) and were mounted on copper slot grids with a specimen support film (Ted Pella Inc., 01805). Sections were stained with uranyl acetate (Electron Microscopy Sciences, 15200) and lead citrate (Electron Microscopy Sciences, 17800). The cells were observed at 120 kV in a Tecnai G2 microscope (Thermo Fisher Scientific, Waltham, MA, USA).

## Heavy and light probes labeling in stably expressed APEX2 cell line for LC-MS/MS analysis

Each stable cell was grown in two T175 flasks. One is for a heavy probe (HDBP or HBP), and the other is for a light probe (LDBP or LBP). Cells were induced with doxycycline at 60–80% confluence. After 18–24 h, the medium in each flask was changed to 15 ml of fresh growth medium containing 250 μM of the probe. The flasks were incubated at 37 °C under 5% $CO_2$ for 30 min according to previously published protocols[6]. Afterwards, 1.5 ml of 10 mM $H_2O_2$ (diluted from 30% $H_2O_2$, Sigma Aldrich H1009) was added to each flask for a final concentration of 1 mM $H_2O_2$, and the flasks were gently agitated for 1 min at room temperature. The reaction was then quenched by adding 15 ml of DPBS containing 10 mM Trolox (Sigma Aldrich, 238813), 20 mM sodium azide (Alfa aesar, 14314), and 20 mM sodium ascorbate (Sigma aldrich, A4034) to each flask. Then, the solution was removed, and the cells were washed two times with a cold quenching solution (DPBS containing 5 mM Trolox, 10 mM sodium azide, and 10 mM sodium ascorbate). Cells were detached using a cold quenching solution and centrifuged at $1000 \times g$ for 2 min at 4 °C. Cells were resuspended with fresh cold quenching solution and centrifuged again. Cells from one T175 flask were lysed with 1.5 ml 2% SDS in 1× TBS (25 mM Tris, 0.15 M NaCl, pH 7.2), 1 × protease inhibitor cocktail (Thermo, 78430), 1 mM phenylmethylsulfonyl fluoride (PMSF) for 5 min at room temperature. Lysates were clarified by ultrasonication (Bioruptor, diagenode) for 15 min with a cold-water bath. 7 ml of cold acetone (−20 °C) was added to each lysate and kept at −20 °C. After at least two h, samples were centrifuged at $13{,}000 \times g$ for 10 min at 4 °C. The supernatant was removed gently. 6.3 ml of cold acetone and 700 μl of 1× TBS were added to the pellet. Samples were vortexed vigorously and kept at −20 °C. After two h to overnight, samples were centrifuged at $13{,}000 \times g$ for 10 min at 4 °C. The supernatant was removed gently. The pellet was allowed to air-dry for 3-5 min and resolubilized with 1 ml of 8 M urea (Sigma Aldrich, U5378) in 50 mM ammonium bicarbonate (ABC, Sigma Aldrich, A6141). The concentration of protein was measured by BCA assay.

## Trypsin digestion and Labeled peptide enrichment

After protein concentration measurement, 6 mg of heavy probe (HDBP or HBP) labeled sample and 6 mg of light probe (LDBP or LBP) labeled sample were mixed. Samples were denatured at 650 rpm for 1 h at 37 °C using a thermomixer (Eppendorf). The samples were reduced by adding dithiothreitol (Sigma Aldrich, 43819) to 10 mM final concentration and incubated at 650 rpm for 1 h at 37 °C using a thermomixer. The samples were alkylated by adding iodoacetamide (Sigma Aldrich, I1149) at a final concentration of 40 mM and mixed at 650 rpm for 1 h at 37 °C using a thermomixer. The samples were diluted eight times using 50 mM ABC. CaCl2 was added to 1 mM final concentration, and 240 μg of Trypsin (Thermo Fisher, 20233) was added to each sample. Samples were incubated at 650 rpm for 6–18 h at 37 °C using a thermomixer. The samples were centrifuged at $10{,}000 \times g$ for 5 min to remove insoluble material. Then, 600 μl of streptavidin beads (Thermofisher Scientific, 88817) were washed with 2 M urea in 1× TBS four times and added to the sample. The samples were rotated for 1 h at

room temperature. The flow-through fraction was kept, and the beads were washed twice with 2 M urea in 50 mM ABC. After removing the supernatant, the beads were washed with pure water in new tubes. Biotinylated peptides were heated at 60 °C and mixed at 650 rpm after adding 300 μl of 80% acetonitrile (Sigma Aldrich, 900667), 0.2% trifluoroacetic acid (Sigma Aldrich, T6508), and 0.1% formic acid (Thermofisher Scientific, 28905). Each supernatant was transferred to new tubes. The elution step was repeated four more times. Combined elution fractions were dried using Speed vac (Eppendorf). Samples can be stored at −20 °C or immediately analyzed by mass spectrometer.

### LC-MS/MS analysis of isotope-coded BP or DBP samples
Analytical capillary columns (100 cm × 75 μm i.d.) and trap columns (2 cm×150 μm i.d) were packed in-house with 3 μm Jupiter C18 particles (Phenomenex, Torrance). The long analytical column was placed in a column heater (Analytical Sales and Services) regulated to a temperature of 45 °C. NanoAcquity UPLC system (Waters, Milford) was operated at a flow rate of 300 nl/min over 2 h with a linear gradient ranging from 95% solvent A (H$_2$O with 0.1% formic acid) to 40% of solvent B (acetonitrile with 0.1% formic acid). The enriched samples were analyzed on an Orbitrap Fusion Lumos mass spectrometer (Thermo Scientific) equipped with an in-house customized nanoelectrospray ion source. Precursor ions were acquired (m/z 300-1500) at 120 K resolving power, and the isolation of precursor for MS/MS analysis was performed with a 1.4 Th. Higher-energy collisional dissociation (HCD) with 30% collision energy was used for sequencing with auto gain control (AGC) target of 1e5. Resolving power for acquired MS2 spectra was set to 30 k at 200 ms maximum injection time.

### MS data processing and protein identification of isotope-coded BP or DBP samples
All MS/MS data were searched by MaxQuant (version 1.6.2.3) with Andromeda search engine at 10 ppm precursor ion mass tolerance against the SwissProt Homo sapiens proteome database (20,199 entries, UniProt (http://www.uniprot.org/)). The MS1-based quantification using either the DBP set (LDBP: + 331.1896 Da; HDBP: + 340.2135 Da) or the BP set (LBP: + 361.1460 Da; HBP: + 369.1729 Da) was performed with the following search parameters: tryptic digestion, fixed carbamidomethylation of cysteine (+ 57.0215 Da), dynamic oxidation of methionine (+ 15.9949 Da), and protein N-terminal acetylation (+ 42.0106 Da). Less than 1% of false discovery rate (FDR) was obtained for unique labeled peptides and unique labeled proteins. For ICS mapping, MS1-based quantified intensity values of reproducible proteins were normalized[69] prior to further analysis. The missing values were filled by imputed values representing a normal distribution around the detection limit. To impute the missing value, first, the intensity distribution of mean and standard deviation was determined, and then for imputation values, a new distribution based on Gaussian distribution with a downshift of 1.8 and width of 0.3 standard deviations was created for the total matrix. All the statistical tests and informatic analysis with data visualization were performed using Perseus (v.1.6.2.3)[40] with implemented WGCNA R-package, InfernoRDN, R studio, and Microsoft excel.

### Tandem mass tag (TMT) labeling
TMT labeling was performed using either TMTpro (Thermo Fisher Scientific, A44522) or TMTproZero (Thermo Fisher Scientific, A44519) reagents. Dried peptide samples were resuspended in 5 μl of 100 mM HEPES (pH 8.5), and 1 μl of TMT reagent (0.5 mg in 25 μl acetonitrile) was added to each sample. The mixture was incubated at room temperature for 1 h. Labeling was repeated by adding an additional 1 μl of reagent. After the reaction, excess reagent was quenched by adding 5% hydroxylamine and incubating for 15 min. Labeled samples were pooled, dried with a SpeedVac (Thermo Fisher Scientific), resuspended in 10 mM ammonium bicarbonate, and desalted on a C18 StageTip.

### LC-MS/MS analysis of Tandem mass tag labeled samples
Analytical capillary columns (100 cm × 75 μm i.d.) and trap columns (2 cm × 150 μm i.d.) were packed in-house with 3 μm Jupiter C18 particles (Phenomenex, Torrance). The long analytical column was placed in a column heater (Analytical Sales and Services) regulated to a temperature of 45 °C. NanoAcquity UPLC system (Waters, Milford) was operated at a flow rate of 300 nL/min over 200 min with a linear gradient ranging from 95% solvent A (H$_2$O with 0.1% formic acid) to 40% of solvent B (acetonitrile with 0.1% formic acid). For SPS-MS3 analysis, an Orbitrap Eclipse Tribrid mass spectrometer (Thermo Fisher Scientific) equipped with an in-house customized nanoelectrospray ion source was used. MS1 spectra were acquired in the Orbitrap over an m/z range of 375–1575 at a resolution of 120 K. Precursors were isolated for MS2 using a 0.7 Th isolation window. Higher-energy collisional dissociation (HCD) was performed with a collision energy of 30%, and a normalized automatic gain control (AGC) target of 100%. MS2 spectra were acquired in the Orbitrap at a resolution of 15 K with a maximum injection time of 60 ms. For MS3, ten synchronous precursor selection (SPS) ions were selected, fragmented using HCD at 65% collision energy, and analyzed in the Orbitrap at a resolution of 30 K. TurboTMT was enabled, and the scan range was set from m/z 100 to 300. The normalized AGC target was set to 500%, and the instrument operated with a cycle time of 2 s.

### MS data processing and protein identification of TMT-labeled samples
Raw files from SPS-MS3 analysis were processed using Proteome Discoverer v.3.1 (Thermo Fisher Scientific) with the SequestHT algorithm. Database searches were performed against a SwissProt human proteome (UP000005640; downloaded March 2020) with the following parameters; a precursor mass tolerance of 10 ppm, fragment mass tolerance of 0.6 Da, and full tryptic digestion allowing up to two missed cleavages. Variable modifications included methionine oxidation (+ 15.9949 Da), protein N-terminal acetylation (+ 42.0106 Da), and tyrosine modification with desthiobiotin-phenol (+ 331.1896 Da). Static modifications were set as TMTpro labeling at lysine residues and peptide N-termini (+ 304.207 Da), and carbamidomethylation of cysteine residues (+ 57.0215 Da). Peptide-spectrum matches were filtered to a 1% false-discovery rate (FDR) using the Percolator algorithm. Reporter ion quantification was further filtered using a co-isolation threshold of 75%, signal-to-noise (S/N) ratio threshold of 10, and an SPS mass matches threshold of 65%.

### Desalting
The samples were desalted using a C18 SPE column. Discovery DSC-18 (Supelco) was activated with 3 ml of methanol followed by 3 ml of water/0.1% trifluoroacetic acid. Then, the sample was loaded on the SPE column for binding. The column was washed using 3 ml of 5% acetonitrile/0.1% trifluoroacetic acid (v/v). Finally, desalted peptides were eluted from the column by adding 1 ml of 80% acetonitrile/0.1% trifluoroacetic acid (v/v). The eluted fraction was dried using a speed-vac.

A home-made Stage-tip was prepared for purification of a small amount of peptides. The end of a 200-μL Eppendorf tip was blocked with a 3 M Empore C8 disk (3 M Bioanalytical Technologies, 2214) and filled with ~5 mg of POROS Oigo R2 reversed-phase resin (Applied Biosystems, 1-1159-06) in methanol. The column was activated by sequential centrifugation at 500 × g with 150 μL of 3 % acetonitrile/0.1% formic acid (v/v), 100 μL of 100 % acetonitrile, and 150 μL of 3 % acetonitrile/0.1 % formic acid (v/v). The peptides in the 0.1% formic acid were then loaded onto the column. The column was washed twice with 200 μL of 5 % acetonitrile/0.1% formic acid (v/v). Peptides elution was performed with 150 μL of 70 % acetonitrile/0.1% formic acid (v/v), followed by 50 μL of 100% acetonitrile. The eluted fraction was dried in a speed-vac and stored in a refrigerator prior to LC-MS analysis.

## Statistics and reproducibility

Statistical analyses were performed using GraphPad prism (v.10.4.2) and excel. The statistical significance was obtained using an unpaired two-tailed *t*-test or Modified Fisher's Exact Test. The *p* values lower than 0.05 ($p < 0.05$) were considered statistically significant, and the following symbols were used: * ($p < 0.05$), ** ($p < 0.01$), *** ($p < 0.001$), and **** ($p < 0.0001$). Western blotting and imaging experiments were performed in three independent replicates to verify reproducibility. Representative images are shown in each figure.

## Reporting summary

Further information on research design is available in the Nature Portfolio Reporting Summary linked to this article.

## Data availability

The proteomics data generated in this study has been deposited in the ProteomeXchange database under accession code PXD039492. All other data supporting the results of this study are available in the article and its supplementary files. Source data are provided with this paper.

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

## Acknowledgements

This work was supported by the National Research Foundation of Korea (NRF) under the following grant numbers: NRF-2022R1A2B5B03001658, NRF-2022M3H9A2096199, RS-2023-00260454 and RS-2023-00265581 to H.W.R., RS-2024-00440824 and RS-2024-00343424 (the Bio&Medical Technology Development Program) to J.-S.K., RS-2022-NR069957 and NRF-2019R1A2C1-85154 to J.-M.K, NRF-2021R1A6A3A01087055 to S.S, NRF-2020R1C1C1013927 to M.-G.K., Korea Health Industry Development Institute (KHIDI) funded by the Ministry of Health & Welfare and Ministry of Science and Information & Communication Technology (ICT), Republic of Korea (grant number: RS-2023-KH136879 to H.W.R.). J.Y.M. was supported by the Korea Brain Research Institute basic research program funded by the Ministry of Science and ICT (22-BR-04-04). H.W.R. was supported by Samsung Science and Technology Foundation (SSTF-BA2201-08 to H.W.R.). J.-S.K. thanks to the support of the Institute for Basic Science (IBS-R008-D1) funded by the Ministry of Science and ICT of Korea and the Creative-Pioneering Researchers Program through Seoul National University. S.-Y.L. was supported by the DGIST Start-up Fund Program of the Ministry of Science and ICT (2025020055). M.-G.K. was supported by the BK21 FOUR (Fostering Outstanding Universities for Research) funded by the Ministry of Education, Korea. The MIC60 antibody was a generous gift from Dr. Joo Hyun Lim (Korea National Institute of Health) and Prof. Sang Ki Park (POSTECH, Korea).

## Author contributions

M.-G.K., S.S., J.-M.K., J.-S.K., and H.W.R. conceived the project. O.K., P.K.M., and J.-M.K synthesized the probe for proximity labelling. M.-G.K., S.S., D.-G.J., and S.-Y.L. prepared the LC-MS/MS samples. S.S., D.-G.J., and J.-S.K. performed LC-MS/MS analysis. M.-G.K. conducted western blotting, confocal imaging, and EM imaging. M.J. and J.Y.M. performed EM and CLEM imaging. M.-G.K., S.S., J.-M.K., J.-S.K., and H.W.R. wrote the manuscript. All authors edited the manuscript.

## Competing interests

A patent application with M.-G.K. and H.W.R. has been filed by Seoul National University relating to this work. The remaining authors declare no competing interests.
