## [Transparent Peer Review file · Nature Communications]

Intracristal space proteome mapping using super-resolution proximity labeling with isotope-coded probes

Corresponding Author: Professor Hyun-Woo Rhee

Version 0:

Reviewer comments:

Reviewer #1

(Remarks to the Author)

In this study, Kang et al. describes ICAX (isotope-coded phenol probes for APEX labeling), a method to compare two APEX labeled cells using light and heavy isotope labeled desthiobiotin phenol. This approach builds on the widely adopted APEX technique to allow comparison of labeled protein targets in the same mass spectrometry experiments. The authors applied it successfully to map the sub-proteome of ICS and OCS, two organelle sub-compartments within the mitochondria that are not amenable to physical separation, and hence the proteome remains unknown. Overall the manuscript is well written, presents interesting data that showcases the method and presents an advance to the field, but could also use some clarification or additional experiments at several places. I have the following suggestions.

1. The methodological advance of the present study is not clearly presented in the way the manuscript is currently written. Because the light and heavy isotopes have to be added to two separate plates of cells, the approach is essentially a minor variant of standard APEX techniques. From what I can tell, the major difference here is that the approach allows two separate labels to be compared in one experiment. This could be done rather easily with SILAC labels in the HEK293 cells used in the study. The authors should present in more details if there are additional methodological refinements or advantages of the method that ICAX presents as applied to different study goals.
2. Another straightforward way to achieve the desired duplexing would be to use tandem mass tag, which labels the free N terminus and lysine ends of peptides. At first glance this would be more accessible to more labs as it doesn't require custom synthesis of a new reagent and moreover can be multiplexed up to 35 channels. The authors suggested that TMT conjugation may not be efficient in proximity labeled peptides, but it is not immediately clear why. One straightforward way to demonstrate this is to take the mitochondrial isolate from MTS-APEX cells with or without BP, then perform lysis and digestion and tag with TMT, and compare the TMT completeness of proximity labeled peptides +/- phenol/
3. The light and heavy labeled desthiobiotin phenol are labeled in two separate sets of experiments. How are the intensity values normalized across experiments to derive the needed proximity ratios, and what are the baseline variability of this system? This would be critical to establish that the approach is useful to eliminate spectrum-to-spectrum variance at the mass spectrometry level. This could be examined by separately using light or heavy BP in the same APEX-tagged protein, then combine for mass spectrometry and analyze the distribution of light/heavy ratios. Moreover, it is not clearly presented how the conversion from ratio to spatial distance is calibrated.
4. The isotopic purity of the light and heavy DBP is critical to successful application of the method especially by other groups. This could perhaps be shown in HBP-only samples, by demonstrating there is no light peak in the chromatogram of mass spectrometry experiments.
5. An advantage of the APEX approach and the digest-prior-to-avidin workflow used here is that only Y-containing modified peptides are captured. It would be valuable therefore to map the identified peptides to the presented ICS/OCS proteome (Figure 3) to show that the peptides conform to the expected topology, e.g., only IMS facing peptides on the IMM are identified. In addition, this could be useful for examining whether the integrity of the IMM is compromised.
6. The manuscript mentioned in a few places that the ICS is "enclosed" because it is not susceptible to pH changes, but that is not clear from the schematic presented or the microscopy data. The authors should clarify what they intend or otherwise investigate whether a membrane enclosed compartment really exists that surround some ICS proteins.
7. The wording "super-resolution" in the title is not clearly defined. Since APEX labels at > 200 nm and multiple copies of the same proteins are tagged, it is not clear whether the spatial resolution of the spatial proteomics techniques reach super resolution as conventionally understood. Similarly, the word "multiplexed" could use additional justification, especially given that the demonstrated multiplexing capacity is quite a bit lower than SILAC (3-6 plex) or TMT (10-35 plex) approaches.

Reviewer #2

(Remarks to the Author)

The inner membrane of mitochondria forms invaginations called cristae. The MICOS complex of the inner membrane is critical for the formation of the cristae and for contacts between the inner and outer membrane. The cristae membranes are enriched for proteins of the OXPHOS complexes and the ATP synthase. The inner membrane translocases (TIM23 and TIM22 complexes) were proposed to be enriched in the boundary membrane and largely absent from the cristae membrane. It was speculated that MICOS actively sorts inner membrane proteins but convincing evidence for this hypothesis was not presented so far. In the present study, Kang et al use a proximity label strategy based on APEX2 to distinguish cristae proteins from inner boundary proteins. To this end, they compared targets of two different IMS proteins, AGK and TMEM177. The latter is supposed to reside in cristae membranes, based on their initial APEX2 labeling experiment. Using this one assay, they present a hypothetical map of inner membrane proteins (Fig. 3C). Moreover, the authors show that some matrix proteins can accumulate in the IMS in the presence of ionophores which prevent mitochondrial import. Finally, the authors propose that the specific location within the IMS influences the folding and oxidation state of proteins, suggesting that the different subdomains have distinct physico-chemical properties.

The proximity labeling approach described here with different isotopic APEX probes is very innovative and interesting. However, it has the caveat that the different reporters are not used in the same cell. Rather, peptides from different cell extracts are mixed. Nevertheless, this is an interesting study with value for the researchers studying mitochondria as well as for scientists using APEX2-based proteomics.

Specific:

#1. The dataset shown in Fig. 3C is a core figure of the study. Where are all the subunits of the respiratory chain and the ATPase? At least many of the proteins of complexes I, III and IV with regions exposed into the IMS would be expected. Their colocalization in the same subdomain should serve as a prove-of-concept for the dataset.

#2. According to Fig. 3C, Tim9 and Tim10 are located in different subcompartments of the IMS? This is surprising as both proteins form a stable 3:3 complex. The authors have to validate their sublocalisation data by another, proximity labeling-independent method.

#3. The MICOS complex is essential for cristae formation. The authors should deplete MICOS components (e.g. by silencing) which should wipe out the differences in their mapping approach with TMEM177 and AGK.

#4. The authors claim that matrix proteins accumulate in the IMS upon addition of an uncoupler. Addition of the uncoupler would also prevent the translocation of the TMEM177-APEX2 reporter into mitochondria. Thus, the APEX2 labeling might simply occur outside of the mitochondria, on the mitochondrial surface or in the cytosol, which could easily explain why some matrix proteins are biotinylated. The biotinylation might simply occur on the surface of non-functional mitochondria. This needs to be excluded and the authors have to show the IMS location of matrix proteins by other methods which do not rely on the targeting of APEX2 or other reporters.

#5. The authors propose that proteins in the different subcompartments of the IMS differ in respect to their thermostability and redox status of cysteines. They speculate that some proteins might be stabilized by disulfide bonds. They even present AlphaFold structures for OMA1, NDUFS5 and NDUFB10. The disulfide bonds in S5 and B10 were already experimentally confirmed in the past. Both proteins are MIA40 substrates (Salscheider et al., 2022, EMBO J 41, e110784). The disulfide bond in OMA1 needs to be experimentally validated. In general, the last section of the study is less convincing. According to Fig. 3C, MIA40 substrates are present in both regions of the IMS. In order to conclude that the conditions in both regions differ, the authors have to show that the redox states of at least one endogenous protein depends on where in the IMS it is localized. The use of a roGFP reporter (Fig. 5c) is not sufficient as this might be simply influenced by the glutathione redox potential. Since Fig. 5 is highly speculative and experimentally a bit 'thin', I suggest to remove Fig. 5 and to give more space to the data shown in Figs. 1 to 4.

Reviewer #3

(Remarks to the Author)

Comments:

Kang et al presented a study in which they developed a new strategy to construct protein distribution maps between two non-partitioned proximal spaces. Their multiplexed proximity labeling approach using isotope-coded phenol probes for APEX labeling (ICAX) enables the quantitative analysis of the spatial proteome at nanometer resolution between two distinctly localized APEX enzymes. Using this technique, they successfully profiled ICS and OCS proteome and corresponding microenvironment. They also presented the dynamic proteome changes in the ICS under conditions of energy stress. Although the probes used were the isotope-coded probe based on commonly-used proximity probe biotin-phenol (BP) probe, the methods of calculating the mass signal intensity ratio between two distinctly localized APEX proteins, leading to the profiling of protein distribution maps between two non-partitioned proximal spaces was quite interesting. There are some questions need to be explained. Questions are listed below.

1. The authors presented the equal reaction efficiency through MS analysis between HBP and LBP. However, they didn't evaluate the reaction efficiency through MS analysis of HDBP and LDBP, which were used in their subsequent experiments. They only presented western blot analysis, which was a semi-quantification method. Could the authors give the equal reaction efficiency through MS analysis between HDBP and LDBP?

2. In extended data figure 2e, the confocal images using streptavidin staining showed diffused fluorescence signal instead of overlap with anti-v5 signal. Is it because of the comparatively low signal intensity? I wonder the comparison of the labeling efficiency of the probes used in this paper with normally used APEX probes.

3. After the author demonstrated TMEM177 and AGK was ICS-localized and OCS-localized respectively, they aimed to reveal more ICS and OCS proteomes using ICAX on TMEM177-APEX2 and AGK-APEX2 stable cell lines. Since they already identified ICS and OCS markers, is it possible to apply existing proximity labeling methods of short labeling radius (few nanometers) instead of ICAX. Compared with these proximity labeling methods of short labeling radius, what are the advantages of ICAX? Could the authors give the data of proximity labeling methods of short labeling radius and compare it with their ICAX method?

4. The authors demonstrated that their ICAX methods could identify ICS proteins and finally identified 4 proteins whose $\log_2 FC_SCO1/TDRKH > 20$ as strong ICS protein candidates. However, they only presented the EM images of one protein. Could the authors give the EM images of the other three proteins so as to justify the reliability of their ICAX approach?

5. The authors classified the ICAX-labeled proteins in the experiment of SCO1-APEX2 and TDRKH-APEX2 into 4 sub-mitochondrial populations based on their $FC_SCO1/TDRKH$ values. They chose proteins of $\log_2 FC_SCO1/TDRKH > 20$ as group 1. Are there any criteria for the chosen of FC value? Why chose 20? In Line 252, the author mentioned that "we selected 64 proteins that were highly enriched by TMEM177-APEX2 (group ICS)". There seems to be no mention of the screening criteria.

6. The author used the TMEM177-APEX2 stable cell line to monitor the changes in ICS proteome under conditions of energy stress. However, if TMEM177 would relocalize, the above conclusions would be unreliable. The authors should present some experimental evidence that TMEM177 would not relocalize under conditions of energy stress.

7. The discussion section was too brief. Could the authors give more perspective and applications of this ICAX method.

8. There might exist some errors in this manuscript. For example, "Fig. 1g" in Line 197 probably be "Fig. 2g" and "Fig. 2f" in Line 213 probably be "Fig. 2g". "pHluorin2" in the last line of images in extended data figure 8a probably be "gTEMP". The authors need to be further checked.

Version 1:

Reviewer comments:

Reviewer #1

(Remarks to the Author)

The authors have addressed my previous comments clearly and convincingly. The investigations on TMT labeling of biotinylated and attempted optimization are excellent and reinforce the value of the approach. The new figures on the topology of biotinylated residues add significantly to the confidence on the specificity of labeling. The manuscript is well written and clearly presented. I congratulate the authors on an excellent study have no problem recommending publication.

Reviewer #2

(Remarks to the Author)

The authors now added a large number of additional data and the MIC60 depletion dataset revealed many novel interesting aspects. This is an excellent study of very high technical quality. The tools developed here will be valuable for many researchers. It is interesting that the authors observed that mitoribosomal proteins can accumulate in the intermembrane space in human cells. This is in agreement with two yeast studies which recently were published and the authors might cite them in the context of matrix proteins that were found in the intermembrane space (PMID: 39890954, 40524011). I now fully support the publication of this study in its present form.

Reviewer #3

(Remarks to the Author)

The authors have addressed all of my previous comments and questions. I have no objection to publication.

REVIEWER COMMENTS

Reviewer #1 (Remarks to the Author):

In this study, Kang et al. describes ICAX (isotope-coded phenol probes for APEX labeling), a method to compare two APEX labeled cells using light and heavy isotope labeled desthiobiotin phenol. This approach builds on the widely adopted APEX technique to allow comparison of labeled protein targets in the same mass spectrometry experiments. The authors applied it successfully to map the sub-proteome of ICS and OCS, two organelle sub-compartments within the mitochondria that are not amenable to physical separation, and hence the proteome remains unknown. Overall the manuscript is well written, presents interesting data that showcases the method and presents an advance to the field, but could also use some clarification or additional experiments at several places. I have the following suggestions.

We sincerely appreciate the reviewer for acknowledging the impact and comprehensiveness of our work. The reviewer provided valuable comments regarding the validation of the performance and advantages of our ICAX method. We have thoroughly addressed each of the major suggestions with supporting experiments, as outlined below. We hope that our revised manuscript will be acceptable to the reviewer.

1. (Reviewer 1-1) The methodological advance of the present study is not clearly presented in the way the manuscript is currently written. Because the light and heavy isotopes have to be added to two separate plates of cells, the approach is essentially a minor variant of standard APEX techniques. From what I can tell, the major difference here is that the approach allows two separate labels to be compared in one experiment. This could be done rather easily with SILAC labels in the HEK293 cells used in the study. The authors should present in more details if there are additional methodological refinements or advantages of the method that ICAX presents as applied to different study goals.

Response: We appreciate the reviewer for raising this critical point. SILAC requires several rounds of cell division for incorporation of isotope-labeled amino acids. Thus, SILAC cannot be applied to various primary cells or human tissues; additionally, its application is hampered by incomplete and cost-ineffective incorporation of isotopes in animal models. Although we used HEK293T cells for proof of concept in this study, recently, various groups successfully applied proximity labeling with APEX2 in animal models, including mice^{1,2}. Thus, we believe that ICAX has methodological advantages over SILAC.

2. (Reviewer 1-2) Another straightforward way to achieve the desired duplexing would be to use tandem mass tag, which labels the free N terminus and lysine ends of peptides. At first glance this would be more accessible to more labs as it doesn't require custom synthesis of a new reagent and moreover can be multiplexed up to 35 channels. The authors suggested that TMT conjugation may not be efficient in proximity labeled peptides, but it is not immediately clear why. One straightforward way to demonstrate this is to take the mitochondrial isolate from MTS-APEX cells with or without BP, then perform lysis and digestion and tag with TMT, and compare the TMT completeness of proximity labeled peptides +/- phenol/

Response: We appreciate your valuable comment. We agree that TMT labeling methods facilitate multiplexed analysis. However, isobaric quantification using TMT often suffers from ratio compression due to co-isolation and co-fragmentation of multiple peptides, which results in underestimation of the actual difference in protein abundance³. Additionally, we conducted several experiments to assess whether the TMT labeling of DBP-modified peptides is inefficient.

In our ICAX analysis, we used an enhanced protocol for identification of biotinylation sites, which was recently developed⁴. This protocol has been carefully optimized for LC-MS analysis, with particular emphasis on minimizing sample loss during the enrichment steps to maximize the identification of biotinylated peptides. A key strategy for reducing sample loss is the direct injection of the eluent into the LC-MS system followed by on-line desalting, bypassing any additional off-line desalting step.

Figure R1. TMT labeling (1:8 peptide-to-TMT ratio) result.

As the reviewer pointed out, we did attempt to label the final enriched biotinylated peptides with isobaric tags, following the manufacturer's protocol for TMT labeling (1:8 peptide-to-TMT ratio). However, the labeling efficiency for biotinylated peptides was only 25.5% (**Figure R1, Extended Figure 15a, group I**). This is possibly due to two major factors. First, the relatively low concentration of DBP-modified peptides (~100 ng/5 μ l, based on the estimation from intensity area) in buffer may lead to TMT hydrolysis rather than effective peptide labeling⁵. Second, the presence of trace contaminants in the eluted peptides may have interfered with TMT labeling. However, in the case of ICAX analysis, such contaminants can be effectively removed by on-line desalting.

Even when using an excess amount of TMT reagent (1:200 peptide-to-TMT ratio) in the labeling reaction, the labeling efficiency was low (~90%) for confidently proceeding to downstream quantitative analysis (**Extended Figure 15a-b, group I**). These observations suggest that TMT labeling of biotinylated peptides in this context is suboptimal and supports our decision to use the ICAX approach for improved identification sensitivity.

Extended Data Figure 15. Comparison between ICAX and isobaric quantification using TMT labeling. (a) Schematic representation of the LC-MS/MS sample procedure for TMT labeling after peptide-level and conventional protein-level enrichment. (b) Bar graphs showing the TMT labeling efficiency in each group. Partially labeled peptides indicate that TMT was labeled at least on the N-terminus or lysine residues. In this triplicate experiment, TMT-zero was used for labeling. Significance values for triplicate experiments were calculated using an unpaired two-tailed t-test. Error bars represent the standard error of the mean. (c) Schematic representation of the LC-MS3 analysis with TMT-labeled samples for LDBP-modified peptides labeled by MTS-APEX2. Peptide desalting was performed for TMT labeling and removal of residual TMT after reaction. (d) Venn diagram showing the number of quantified proteins using TMT labeling ($n = 6$) compared with the ICAX ($n = 3$) results.

Additionally, we further attempted to optimize the isobaric labeling of the enriched biotinylated peptides (Extended Figure 15a, group II). In this approach, the eluted samples were first desalted prior to TMT labeling. This modification led to improved TMT labeling efficiency (Extended Figure 15b, Group II), which is comparable to the conventional TMT protocol for proximity labeling (Extended Figure 15b, Group III). However, it came at a significant cost: the number of identified biotinylated peptides and their corresponding proteins was markedly reduced in the TMT experiment using MTS-APEX2 (Extended Figure 15c, d). We attribute this loss to the multiple desalting steps (e.g., Stage-Tip), which likely introduced additional sample loss. These

findings underscore a critical trade-off between labeling efficiency and peptide recovery, further justifying our choice to adopt an ICAX strategy that better preserves sample integrity and maximizes the identification of biotinylated species.

Despite the aforementioned issues, combining ICAX probes (LDBP and HDBP) with TMT is intriguing, as it effectively expands the multiplexing capability. Further optimization will be required for future applications. We have included these points in the Discussion section.

3. (Reviewer 1-3) The light and heavy labeled desthiobiotin phenol are labeled in two separate sets of experiments. How are the intensity values normalized across experiments to derive the needed proximity ratios, and what are the baseline variability of this system? This would be critical to establish that the approach is useful to eliminate spectrum-to-spectrum variance at the mass spectrometry level. This could be examined by separately using light or heavy BP in the same APEX-tagged protein, then combine for mass spectrometry and analyze the distribution of light/heavy ratios. Moreover, it is not clearly presented how the conversion from ratio to spatial distance is calibrated.

Response: We thank the reviewer for raising an important point regarding the normalization of intensity values and the evaluation of baseline variability in our proximity labeling strategy using isotopically distinct desthiobiotin-phenol (LDBP and HDBP). As noted by the Reviewer, normalization of MS intensity across the samples is essential to eliminate spectrum-to-spectrum variance inherent to label free quantification method. In the ICAX workflow, HDBP- and LDBP-labeled proteins are combined at the lysate level and subsequently processed through an identical analytical pipeline, including sample preparation, chromatographic separation, ionization, and mass spectrometric detection. This design ensures that both isotopically distinct species experience equivalent experimental conditions, thereby enabling accurate relative quantification.

To validate this approach, we first conducted a proof-of-concept experiment using cell lines stably expressing MTS-APEX2 to examine whether both probes have equal reactivity to the APEX enzyme. Cells were treated with an equimolar pre-mixed solution of LDBP and HDBP to ensure simultaneous and competitive access of both probes to proximal protein targets under identical enzymatic and cellular conditions (**Figure 1c**). As a result, we found that log₂-transformed intensity of LDBP-modified peptides showed excellent correlation with that of HDBP-modified peptides (**Figure 1d**), indicating LDBP and HDBP have the same reactivity to APEX2.

Figure 1. Chemical synthesis of isotope-coded ascorbate peroxidase probes for quantitative proximity labeling. (c) Schematic representation of the experimental design of the equal activity test of desthiobiotin-conjugated APEX probes (light desthiobiotin-phenol [LDBP] and heavy desthiobiotin-phenol [HDBP], 50:50 ratio mixture) on cells expressing mitochondrial targeting sequence (MTS)-APEX2. (d) Scatter plot showing MS1 intensity of both LDBP- and HDBP-modified proteins labeled using MTS-APEX2 with equimolar pre-mixed probe solutions. R-squared (R^2) values and trendline slopes were used to validate accuracy. (e) Experimental procedure of duplexed quantification using the ICAX method. (f) Annotated MS/MS spectra of the LDBP- and HDBP-modified peptides (PGY*AAIQAALSSR) for the PDXR protein. (g) Scatter plot showing the MS1 intensity of LDBP- and HDBP-modified peptides identified by ICAX analysis. R^2 values and trendline slopes in triplicate experiments are listed in the **Extended Data Fig. 3a**.

Next, we prepared either LDBP or HDBP labeled samples with MTS-APEX2 expressing cell lines that underwent independent labeling reactions followed by mixing of the lysates; then we trypsinized them prior to peptide-level enrichment. This design allows us to assess technical variability and labeling consistency across replicates (**Figure 1e**). We confirmed that LDBP- and HDBP-modified peptides were well identified in each of the corresponding MS/MS spectrums (**Figure 1f**). The resulting scatter plots of \log_2 -transformed biotinylated peptide intensities demonstrate excellent correlation between replicates (**Figure 1g and Extended Data Figure 3a**), indicating minimal technical variation between light and heavy channels. These results confirm that the ICAX method effectively controls spectrum-to-spectrum variation and supports the robustness of the approach for quantitative proximity profiling.

Extended Data Figure 3. (a) ICAX was conducted by single analysis with a mixture of LDBP- and HDBP-modified peptides by MTS-APEX2 after enrichment. Scatter plot is based on MS1 intensity. R-squared (R²) values and slopes of trendlines are displayed for validation of accuracy and reproducibility across triplicate experiments.

Regarding the conversion of proximity ratios to spatial distances, we agree with the reviewer that this is a complex relationship, not explicitly addressed by our work. At this stage, we focused on comparing the relative proximity between two distinct spaces (e.g., ICS and OCS) rather than their absolute distance. The high reproducibility and linearity of intensity ratios validate the utility of the platforms for comparative proximity mapping. Although it is beyond the scope of the present study, future work incorporating structural modeling or empirical calibration against known distances could enable the conversion of ratio into spatial constraints.

4. (Reviewer 1-4) The isotopic purity of the light and heavy DBP is critical to successful application of the method especially by other groups. This could perhaps be shown in HBP-only samples, by demonstrating there is no light peak in the chromatogram of mass spectrometry experiments.

Response: We sincerely appreciate the reviewer's insightful comment regarding the importance of isotopic purity, so that other researchers can successfully apply our method in their experimental setups.

To evaluate the isotopic purity of the probes, we directly analyzed either LDBP or HDBP probes through LC-MS. Our analysis confirmed minimal cross-contamination between the isotopic species. Especially based on peak area ratios, we detected only 0.007% HDBP in the LDBP solution and 0.202% LDBP in the HDBP solution (**Extended Data Figure 2a**). These values indicate minimal isotopic overlap and ensure that both LDBP and HDBP probes are of high purity, supporting their suitability for accurate and reliable quantitative analysis in our ICAX workflow.

To further assess whether this minor isotopic contamination might affect downstream analysis, we performed an additional experiment using Flp-in T-REx 293 cells stably expressing MTS-APEX2, in which we labeled them separately with either LDBP or HDBP. Across three biological replicates

Extended Data Figure 2. Isotopic purity of the synthesized LDBP and HDBP probes. (a) LC-MS analysis of LDBP and HDBP probes after direct injection. Peak intensity and area were used to calculate isotopic purity. (d) Bar graph of the numbers of LDBP- and HDBP-modified peptides identified in either LDBP- or HDBP-treated samples after peptide-level enrichment. Significance values for triplicate experiments were calculated using an unpaired two-tailed t-test. Error bars represent the standard error of the mean.

of each experiment, we observed that in the LDBP-labeled samples, only ~0.6% of the peptides (8 of 1306 on average) were detected as HDBP-modified. Conversely, in the HDBP-labeled samples, ~0.5% of the peptides (5.6 of 1112 on average) were detected as LDBP-modified (Extended Data Figure 2d). Considering the 1% FDR level on identification and high purity of the reagent, these low levels of isotopic crossover are within the expected range and do not significantly impact the accuracy or interpretation of our ICAX quantification. Therefore, we concluded that the observed background signal is negligible and does not compromise the robustness or reproducibility of our quantitative proximity labeling analysis.

5. (Reviewer 1-5) An advantage of the APEX approach and the digest-prior-to-avidin workflow used here is that only Y-containing modified peptides are captured. It would be valuable therefore to map the identified peptides to the presented ICS/OCS proteome (Figure 3) to show that the peptides conform to the expected topology, e.g., only IMS facing peptides on the IMM are identified. In addition, this could be useful for examining whether the integrity of the IMM is compromised.

Response: We thank the reviewer for this comment. We have updated the topological annotation of modification sites on labeled proteins in our ICS/OCS proteome dataset. In this dataset, we identified 76 IMM proteins with a total of 145 modification sites as defined by the MitoCarta 3.0 database⁶ and literature studies^{7,8}. We compared our modification-site information to determined membrane domains using Uniprot database⁹, TMHMM¹⁰, TMbed¹¹, and AlphaFold3 structure¹². Based on this analysis, we found that 141 DBP-modified sites are exposed to the IMS, whereas four DBP-modified sites face the mitochondrial matrix. Notably, COX5B and ATP5F1B, both of which face the matrix, were identified in our ICS dataset, suggesting that a small fraction of DBP radicals may have reached the via certain transmembrane proteins in the ICS. Furthermore, while the majority of modified sites on IMMT (Y95, Y358, Y543, Y626, Y636) and SLC25A3 (Y141, Y165, Y246) are exposed to the IMS, IMMT (Y33), SLC25A3 (Y196) face the matrix. These findings indicate that only a limited number of DBP radicals diffused into the matrix. We have incorporated this topological information into **Extended Data Figures 10 and 11**.

Extended Data Figure 10. Membrane topology of mitochondrial inner membrane-oriented proteins observed in the ICS group.

Extended Data Figure 11. (a, b) Membrane topology of mitochondrial inner membrane-oriented proteins observed in the OCS group (a) and OMM/cytosol group (b).

Moreover, we prepared stable MIC60 (subunit of MICOS complex) knockdown (KD) cell lines expressing TMEM177 using shRNAs to investigate changes in ICS proteome upon cristae structure disruption. Previous studies have shown that MIC60 depletion leads to the loss of cristae-junction and the formation of aberrant ICS structures¹³. To assess the ICS proteome alteration, we compared the intensity of proteins labeled by TMEM177-APEX2 in MIC60 KD cells with that in control cells.

Figure 4. Proteome mapping in the aberrant ICS induced by the MICOS complex inhibition. (d) Scatter plot showing the log₂-fold change of the identified proteins in shMIC60 compared with shControl cells. Identified proteins were highlighted according to their localization and the modification sites in the mitochondria. **(e)** Proposed model demonstrating that the inhibition of MICOS complex leads to disintegration of membrane structures in aberrant ICS.

We observed a reduced labeling intensity for most IMM proteins with DBP modifications on IMS-side tyrosine residue in MIC60 KD cells. This reduction likely reflects the disconnection of aberrant ICS regions from the OCS, a main place for mitochondrial protein imports, leading to impaired IMM protein turnover. Unexpectedly, we identified 23 mitochondrial matrix proteins and 12 IMM proteins with matrix-side DBP modification (**Extended Data Figure 13b**) that exhibited strong labeling in MIC60 KD cells (**Figure 4d**). Because IMM is generally impermeable to small molecules, these results suggest that IMM integrity was compromised under MICOS complex inhibition, allowing increased permeability to DBP radicals (**Figure 4e**).

Extended Data Figure 13. (b) Membrane topology of IMM proteins showing matrix-side modification by diffused DBP from ICS under MICOS complex inhibition.

6. (Reviewer 1-6) The manuscript mentioned in a few places that the ICS is "enclosed" because it is not susceptible to pH changes, but that is not clear from the schematic presented or the microscopy data. The authors should clarify what they intend or otherwise investigate whether a membrane enclosed compartment really exists that surround some ICS proteins.

Response: We apologize for the ambiguous expression. In the previous version of our manuscript, we initially described our hypothetical observation according to the pH titration results (**Extended Data Figure 12b**). The ratio of TMEM177-pHluorin2 barely changed when we treated buffer solutions without digitonin treatment; thus, we reasoned that ICS is the enclosed space compared with the other sub-compartments.

In the revised manuscript, we have summarized the results of fluorescent sensors for focusing on the proteome mapping results, as suggested by the Reviewer 2. Hence, we have removed the expression "enclosed" in our revised manuscript.

Extended Data Figure 12. Assessment of intracristal and outer intracristal subdomain microenvironments and their correlations with the structural properties of their corresponding proteomes. (b) Titration curves for each pHluorin2 construct, with and without digitonin treatment (n = 6 per group).

7. (Reviewer 1-7) The wording "super-resolution" in the title is not clearly defined. Since APEX labels at > 200 nm and multiple copies of the same proteins are tagged, it is not clear whether the spatial resolution of the spatial proteomics techniques reach super resolution as conventionally understood. Similarly, the word "multiplexed" could use additional justification, especially given

that the demonstrated multiplexing capacity is quite a bit lower than SILAC (3-6 plex) or TMT (10-35 plex) approaches.

Response: We thank the Reviewer for this comment. In the previous study, the super-resolution proximity labeling (SR-PL) was defined as direct detection of biotinylated proteins at the single modified amino acid residue-level^{4, 14}, such as tyrosine residue for APEX2. In the conventional PL method, we cannot examine whether identified proteins are biotinylated because this method analyzes only unmodified peptides after on-bead or in-gel digestion. Thus, non-specifically bound proteins cannot be excluded using the conventional PL methods. By contrast, SR-PL facilitates detecting biotinylated proteins specifically. The wording “super-resolution PL” is analogous to the “super-resolution microscopy,” which can detect up to single molecules, whereas conventional confocal microscopy does not.

We fully agree that the wording “multiplexed” is not appropriate for the current ICAX analysis. We have rephrased the word “multiplexed” to “duplexed.” Thank you for these suggestions!

Reviewer #2 (Remarks to the Author):

The inner membrane of mitochondria forms invaginations called cristae. The MICOS complex of the inner membrane is critical for the formation of the cristae and for contacts between the inner and outer membrane. The cristae membranes are enriched for proteins of the OXPHOS complexes and the ATP synthase. The inner membrane translocases (TIM23 and TIM22 complexes) were proposed to be enriched in the boundary membrane and largely absent from the cristae membrane. It was speculated that MICOS actively sorts inner membrane proteins but convincing evidence for this hypothesis was not presented so far. In the present study, Kang et al use a proximity label strategy based on APEX2 to distinguish cristae proteins from inner boundary proteins. To this end, they compared targets of two different IMS proteins, AGK and TMEM177. The latter is supposed to reside in cristae membranes, based on their initial APEX2 labeling experiment. Using this one assay, they present a hypothetical map of inner membrane proteins (Fig. 3C). Moreover, the authors show that some matrix proteins can accumulate in the IMS in the presence of ionophores which prevent mitochondrial import. Finally, the authors propose that the specific location within the IMS influences the folding and oxidation state of proteins, suggesting that the different subdomains have distinct physico-chemical properties.

The proximity labeling approach described here with different isotopic APEX probes is very innovative and interesting. However, it has the caveat that the different reporters are not used in the same cell. Rather, peptides from different cell extracts are mixed. Nevertheless, this is an interesting study with value for the researchers studying mitochondria as well as for scientists using APEX2-based proteomics.

We sincerely appreciate the reviewer for the insightful comments on our ICS and OCS mapping results, as well as further suggestions, which have been very helpful in improving our work. We conducted additional experiments in an effort to thoroughly address and incorporate the reviewer's suggestion. We hope that our revised manuscript with additional results satisfactorily addresses the reviewer's concerns.

Specific:

1. (Reviewer 2-1) The dataset shown in Fig. 3C is a core figure of the study. Where are all the subunits of the respiratory chain and the ATPase? At least many of the proteins of complexes I, III and IV with regions exposed into the IMS would be expected. Their colocalization in the same subdomain should serve as a prove-of-concept for the dataset.

Response: We appreciate the Reviewer’s insightful comment. We identified 74 IMM proteins using ICAX analysis with TMEM177-/AGK-APEX2. Among them, 16 were subunits of the OXPHOS complexes I, III, IV, and V (**Extended Data Figure 9d**). We found that DBP-labeled sites of 14 IMM proteins were exposed to the IMS; however, COX5B and ATP5F1B facing the mitochondrial matrix were identified. We investigated the membrane topology of 74 IMM proteins in our dataset (please see **Extended Data Figure 10** and **11** in the manuscript) and found that at DBP-labeled sites, 72 IMM proteins (except for COX5B and ATP5F1B) were exposed to the IMS, indicating that a small fraction of DBP radicals moved to the matrix through transmembrane proteins in the ICS.

Among the 16 identified OXPHOS-complex subunits, five proteins (NDUFA8, COX6B1, and NDUFA4, UQCRCQ, CYC1) were classified in the OCS group according to our criteria. We sub-grouped our dataset according to the fold change of OPA1 (Log₂ FC: 0.71) and YME1L1 (Log₂ FC: 0.62), which may localize to the outermost part of ICS space (cristae junction) for the regulation of cristae structures. Although three subunits were sub-grouped as OCS, their log₂-transformed fold change values (COX6B1: 0.45, NDUFA8: 0.44, and NDUFA4: 0.41) were not much lower than those of OPA1 and YME1L1 (**Supplementary Dataset 7**).

Extended Data Figure 9. (d) The scatter plot shows the distribution of subunits in OXPHOS complexes I, III, IV, and V. Protein modification sites exposed to the mitochondrial matrix and IMS are marked in red and blue, respectively.

However, complex III subunits were identified in the OCS. This conflict is possibly due to a characteristic of the APEX-based ICAX approach, which specifically labels the surface-exposed tyrosine residues, potentially resulting in insufficient labeling of structurally obscured complexes. Since ICS proteins must transit through the OCS during the import process, it is possible that certain complex III subunits are transiently retained in the OCS. During this retention, their tyrosine residues may become more surface-exposed compared to assembled complex, facilitating

their modification by AGK-APEX2. To address this issue, complementary PL methods capable of labeling different amino acid residues should be employed to validate our findings on ICS and OCS proteomes in the future. We have included these points in the Discussion section.

2. (Reviewer 2-2) According to Fig. 3C, Tim9 and Tim10 are located in different subcompartments of the IMS? This is surprising as both proteins form a stable 3:3 complex. The authors have to validate their sublocalisation data by another, proximity labeling-independent method.

Response: We appreciate that the Reviewer raised this critical issue. We agree and respect previous findings that TIM9 and TIM10 form a hetero-hexameric complex in the OCS for mitochondrial protein import. Thus, we anticipated that TIM9 would be co-localized with TIM10 based on imaging analysis. Nonetheless, we have provided an explanation for the differential localization of TIM9 and TIM10 observed in our dataset. In the TIM9-TIM10 complex, the surface-exposed tyrosine residues of TIM9 and TIM10 are mutually masked upon assembly. Therefore, unassembled TIM10 is more likely to be labeled by TMEM177-APEX2 than its complex form.

To address this point, we conducted isobaric quantification using TMEM177- and AGK-APEX2 following conventional protein-level enrichment (**Figure R2a**), a method that may facilitate the analysis of protein complexes. As a result, we identified 2258 proteins including 137 mitochondrial matrix proteins, 164 IMM, 28 IMS, 61 OMM proteins⁶ (**Figure R2b**). Compared to ICAX dataset, this approach yielded a high proportion of non-specific binding proteins, likely due to analysis of unmodified peptides via on-bead digestion, indicating that ICAX provides a more specific result. Nonetheless, we selected 253 proteins including IMM, IMS, and OMM proteins for further analysis.

We classified the ICS and OCS group based on the log₂-transformed FC values, consequently, we found that TIM10 was identified in the ICS (**Figure R2c**). This finding suggests the presence of monomeric or unassembled TIM10 pool in the ICS. This may indicate a role for local quality control or regulation of TIM subunits in ICS. Unassembled TIM10 from TIM9 are known to be degraded by Yme1¹⁵, whose human homolog (YME1L1) was found in the ICS in current study. We have described these points in the Discussion section.

Figure R2. (a) Schematic representation of isobaric quantification using TMT labeling following a conventional protein-level enrichment process. (b, c) LC-MS³ results from isobaric analysis of TMEM177-APEX2 and AGK-APEX2 for mapping ICS and OCS proteomes. A heat map displaying the total identified proteins (b) and mitochondrial IMM, IMS, OMM proteins (c) based on the FC values. TIM10 is indicated with an arrow in (c).

3. (Reviewer 2-3) The MICOS complex is essential for cristae formation. The authors should deplete MICOS components (e.g. by silencing) which should wipe out the differences in their mapping approach with TMEM177 and AGK.

Response: We appreciate the Reviewer's valuable comment. The mitochondrial structure in MIC60-deficient cells has been well characterized by EM imaging in various studies^{13, 16}. Recently, Jakobs group demonstrated that mitochondria in MIC60-knockout (KO) cells exhibit a very low frequency of cristae junctions, leading to aberrant cristae structures¹³. Based on these findings, we generated control and MIC60-knockdown (KD) cell lines expressing TMEM177-APEX2 using the shRNA to examine changes in the ICS proteome under inhibition of the MICOS complex. We compared MIC60 KD cells expressing TMEM177-APEX2 with the control cells, but not MIC60 KD cells expressing AGK-APEX2, in order to avoid variability arising from knockdown in different cell lines.

We validated the biotinylation pattern of TMEM177-APEX2 in shControl and shMIC60 cells using western blotting and confocal imaging. We observed mitochondrial aggregation in MIC60 KD cells, consistent with previous reports in MIC60 KO mice¹⁶. The TMEM177-APEX2 biotinylation in MIC60 KD cells was restricted rather than diffusive pattern, suggesting that DBP radicals were confined within the aberrant ICS (**Figure 4a, b**). Additionally, some biotinylation bands in MIC60-KD cells were slightly reduced compared with the control cells, which may be owed to the restricted diffusion of the labeling (**Figure 4c**).

Then, we conducted ICAX analysis for shControl and shMIC60 cells using LDBP and HDBP probes and identified 221 proteins, including 33 matrix proteins, 91 IMM proteins, and 15 IMS proteins. We found that subunits of TIM and MICOS complex, labeled in the IMS, and OMM proteins, labeled in the cytosol, were enriched in the control cells (**Figure 4d**). We also matched our OCS dataset, consequently confirming that OCS proteins were enriched in the control cells (**Extended Data Fig. 13a**). These results indicate that DBP radicals generated by TMEM177-APEX2 were not accessible in the OCS and cytosol owing to the formation of an aberrant ICS. The biogenesis of the OXPHOS complex is highly dependent on TIM23 complex, which directly inserts most of the subunits laterally. However, this process can be disrupted in aberrant ICS due to a disconnection with the TIM23 complex at the inner boundary membrane. In this context, we found that several subunits of ICS-localized OXPHOS complexes showed reduced labeling intensity in MIC60 KD cells (**Extended Data Fig. 13a**), which corresponds to the attenuated OCR observed in MIC60 deficient cells¹³.

Figure 4. Proteome mapping in the aberrant ICS induced by the MICOS complex inhibition. (a) Confocal imaging results showed aggregated mitochondria in the MIC60 KD cells. Anti-V5 and anti-TOM20 antibodies were used to visualize TMEM177-APEX2 and mitochondria. Scale bars represent 10 μ m. (b) Line scan analysis for ROI in (a) shows diffusivity of DBP radicals. (c) Biotinylation level in MIC60 KD cells was detected by western blotting using streptavidin-HRP. TMEM177-V5-APEX2 expression level was visualized using an anti-V5 antibody, and GAPDH was used as a loading control. Reduced expression level of MIC60 in KD cells were determined by anti-MIC60 antibody. (d) Scatter plot showing the log₂-fold change of the identified proteins in shMIC60 compared with shControl cells. Identified proteins were highlighted according to their localization and the modification sites in the mitochondria. (e) Proposed model demonstrating that the inhibition of MICOS complex leads to disintegration of membrane structures in aberrant ICS.

Surprisingly, we found that 33 mitochondrial matrix proteins and 18 IMM proteins, labeled on the matrix-side, were enriched in shMIC60 cells (Figure 4d), similar to the BAM15 results (Figure 5). However, we believe that the mechanism of diffusion differs in these stress conditions and the

number of identified matrix proteins in MIC60-KD cells are much fewer than that under BAM15-treated conditions. MIC60 KD induces aberrant ICS formation, resulting in its confinement from the OCS which is a key place for mitochondrial protein import. IMM is generally impermeable, as it is enriched with selective transporters and cardiolipin to maintain metabolism. However, the integrity of aberrant cristae membranes may be compromised by unregulated protein biogenesis potentially allowing DBP radical penetration. By contrast, DBP radicals may pass through the mitochondrial permeability transition pore (mPTP), which is induced by the mitochondrial uncoupling¹⁷⁻¹⁹ conditions (please see the response to the following comment for further detail).

Overall, these findings raise interesting questions regarding the regulation of membrane permeability within the ICS under various stress conditions. We have included these observations in the revised manuscript.

a

Extended Data Figure 13. (a) Scatter plot showing the list of ICS and OCS proteome identified in TMEM177-, and AGK-APEX2 dataset. OXPHOS subunits with IMS-side DBP modification are shown in the right panel.

4. (Reviewer 2-4) The authors claim that matrix proteins accumulate in the IMS upon addition of an uncoupler. Addition of the uncoupler would also prevent the translocation of the TMEM177-APEX2 reporter into mitochondria. Thus, the APEX2 labeling might simply occur outside of the mitochondria, on the mitochondrial surface or in the cytosol, which could easily explain why some matrix proteins are biotinylated. The biotinylation might simply occur on the surface of non-functional mitochondria. This needs to be excluded and the authors have to show the IMS location of matrix proteins by other methods which do not rely on the targeting of APEX2 or other reporters.

Response: We thank the Reviewer for raising this critical point. We agree that accumulated mitochondrial proteins can be labeled by non-imported TMEM177-APEX2 in the cytosol upon addition of BAM15. Moreover, a small fraction of DBP radicals generated by TMEM177-APEX2 can diffuse to the cytosol through transmembrane proteins (porins) on the outer membrane of mitochondria. Thus, we used a mitochondria fractionation method and protein synthesis inhibitors to rule out that TMEM177-APEX2 targets and labels proteins in the cytosol.

We found that HSPD1 accumulates more in the cytosol fraction after mitochondria isolation (**Extended Data Figure 14d**), so we conducted enrichment of biotinylated proteins in the mitochondria fraction using streptavidin-conjugated beads. We found that biotinylated HSPD1 was well enriched by BAM15 treatment, indicating that HSPD1 was biotinylated in the mitochondria rather than the cytosol. Moreover, the biotinylation signal was higher in the mitochondrial fraction following BAM15 treatment, indicating that other mitochondrial matrix proteins were labeled by TMEM177-APEX2 in the mitochondria (**Extended Data Figure 14d**).

Next, we used puromycin and cycloheximide (CHX) to prevent the accumulation of TMEM177-APEX2 and mitochondrial matrix proteins in the cytosol. To monitor protein synthesis inhibition, we pre-treated cells with 35 μ M of puromycin and CHX for 2 h and added azidohomoalanine (AHA) for 30 min. AHA is a methionine derivative, so we used AHA with methionine-free media to allow its incorporation. Then, newly synthesized proteins were monitored by click reaction between AHA and Cy5-alkyne. We found that protein synthesis was efficiently blocked by puromycin and CHX (**Extended Data Figure 14e**). Next, we pre-treated cells with 35 μ M of puromycin and CHX for 2 h and added BAM15 and LDBP for labeling to determine whether

HSPD1 is labeled by TMEM177-APEX2 in the mitochondria. Then, we found that the HSPD1 biotinylation level was not changed by puromycin and CHX treatment (**Extended Data Figure 14f**). These results indicate that biotinylation of mitochondrial matrix proteins by TMEM177-APEX2 occurs in the mitochondria under mitochondrial uncoupling conditions.

In addition, we conducted further experiments to examine the sub-compartment where the DBP-modifications of mitochondrial matrix proteins occur. We expressed MTS-dsRed in the TMEM177-APEX2 stable cells, and biotinylated MTS-dsRed was enriched using streptavidin-conjugated beads. As a result, we found that the processed form of MTS-dsRed, cleaved by mitochondrial processing peptidase, was enriched under BAM15 treatment (**Extended Data Figure 14i**). In contrast, only the unprocessed form of MTS-dsRed was biotinylated by TMEM177-APEX2 under the normal condition, indicating that mitochondrial matrix proteins were labeled specifically within the matrix. We believe that mPTP opening is involved in this process, as it has been reported that mPTP is induced by mitochondrial uncoupling¹⁷⁻¹⁹.

Overall, our collective data demonstrated that DBP radicals generated from ICS crossed the IMM through the mPTP and modified the mitochondrial matrix proteins under mitochondrial uncoupling conditions.

d

e

f

i

Extended Data Figure 14. Validation of mitochondrial matrix proteins labeled with TMEM177-APEX2 under BAM15 treatment. (d) Detection of biotinylated proteins labeled by TMEM177-APEX2 in isolated mitochondria. (e) Measurement of the protein synthesis level using L-azidohomoalanine (AHA) after puromycin and cycloheximide (CHX) treatment. AHA-incorporated proteins were labeled with Cy5-alkyne by click reaction for detection. (f) Enrichment of biotinylated proteins labeled by TMEM177-APEX2 using streptavidin-conjugated beads under the mitochondrial uncoupling. Cells were treated with puromycin and CHX for 2 h followed by addition of BAM15 and LDBP for 1 h. (i) Western blot shows that the processed form of MTS-dsRed that was strongly biotinylated by TMEM177-APEX2 under mitochondrial uncoupling.

5. (Reviewer 2-5) The authors propose that proteins in the different subcompartments of the IMS differ in respect to their thermostability and redox status of cysteines. They speculate that some proteins might be stabilized by disulfide bonds. They even present AlphaFold structures for OMA1, NDUFS5 and NDUFB10. The disulfide bonds in S5 and B10 were already experimentally confirmed in the past. Both proteins are MIA40 substrates (Salscheider et al., 2022, EMBO J 41, e110784). The disulfide bond in OMA1 needs to be experimentally validated. In general, the last section of the study is less convincing. According to Fig. 3C, MIA40 substrates are present in both regions of the IMS. In order to conclude that the conditions in both regions differ, the authors have to show that the redox states of at least one endogenous protein depends on where in the IMS it is localized. The use of a roGFP reporter (Fig. 5c) is not sufficient as this might be simply influenced by the glutathione redox potential. Since Fig. 5 is highly speculative and experimentally a bit ‘thin’, I suggest to remove Fig. 5 and to give more space to the data shown in Figs. 1 to 4.

Response: We appreciate the Reviewer’s valuable comment. We have removed the results about disulfide bonds with AlphaFold structures. Furthermore, we have summarized and moved the results of pHluorin2 and gTEMP fluorescent sensors from **Figure 5** to **Extended Data Figure 12**. Because we agree that the roGFP reporter is highly influenced by the glutathione redox potential, we generated stable cell lines expressing mitochondria-targeted HyPer7, which is a sensitive indicator for hydrogen peroxide²⁰. We confirmed the localization of AGK-, TMEM177-, and TOM20-HyPer7 in the mitochondria (**Extended Data Figure 12a**). Using these constructs, we found that the OCS had a comparatively oxidized environment compared to the ICS (**Extended Data Fig. 12d**). This result corroborates with our finding on CHCHD4 in OCS, which is known to form multiple disulfide bonds under physiological conditions involved in oxidative folding system²¹. We have included these results in the “ICS and OCS proteome mapping using ICAX” section.

Extended Data Figure 12. (a) Confocal images of various fluorescent-protein, sensor-conjugated TMEM177 and AGK constructs. All scale bars represent 10 μ m. (d) The redox state was measured with HyPer7 targeting the OCS, ICS, and cytosol-exposed OMM (n = 10 per group).

Reviewer #3 (Remarks to the Author):

Comments:

Kang et al presented a study in which they developed a new strategy to construct protein distribution maps between two non-partitioned proximal spaces. Their multiplexed proximity labeling approach using isotope-coded phenol probes for APEX labeling (ICAX) enables the quantitative analysis of the spatial proteome at nanometer resolution between two distinctly localized APEX enzymes. Using this technique, they successfully profiled ICS and OCS proteome and corresponding microenvironment. They also presented the dynamic proteome changes in the ICS under conditions of energy stress. Although the probes used were the isotope-coded probe based on commonly-used proximity probe biotin-phenol (BP) probe, the methods of calculating the mass signal intensity ratio between two distinctly localized APEX proteins, leading to the profiling of protein distribution maps between two non-partitioned proximal spaces was quite interesting. There are some questions need to be explained. Questions are listed below.

We sincerely appreciate the reviewer's valuable comments regarding the validation of our ICAX method through proof-of-concept experiments and comparison with other proximity labeling methods. In response to this recommendation, we conducted several additional experiments and revised manuscript accordingly. We hope that our revised manuscript, including the new results, adequately addresses the reviewer's concern.

1. (Reviewer 3-1) The authors presented the equal reaction efficiency through MS analysis between HBP and LBP. However, they didn't evaluate the reaction efficiency through MS analysis of HDBP and LDBP, which were used in their subsequent experiments. They only presented westernblot analysis, which was a semi-quantification method. Could the authors give the equal reaction efficiency through MS analysis between HDBP and LDBP?

Response: We thank the reviewer for raising an important point regarding the reaction reactivity of isotopically distinct desthiobiotin-phenol (LDBP and HDBP) used in our study.

To address this, we performed proof-of-concept experiment using MTS-APEX2-expressing cell lines to examine whether both probes exhibit equal reactivity with the APEX enzyme. Cells were treated with an equimolar pre-mixed solution of LDBP and HDBP to ensure simultaneous and competitive access of both probes to proximal protein targets under identical enzymatic and cellular conditions (**Figure 1c**). We found that the log₂-transformed intensity of LDBP-modified peptides showed excellent correlation with that of HDBP-modified peptides (**Figure 1d**), indicating LDBP and HDBP have the same reactivity with APEX2. This result demonstrates that both probes react with comparable efficiencies under identical experimental conditions. Moreover, additional replicates consistently showed high reproducibility between HDBP and LDBP labeling

(Figure R3), further supporting their equivalent labeling performance. These quantitative LC-MS data provide a more robust evaluation than western blot alone and confirm that LDBP and HDBP can be used interchangeably in our multiplexed ICAX workflow without introducing systematic bias.

Figure 1. Chemical synthesis of isotope-coded ascorbate peroxidase probes for quantitative proximity labeling. (c) Schematic representation of the experimental design of the equal activity test of desthiobiotin-conjugated APEX probes (light desthiobiotin-phenol [LDBP] and heavy desthiobiotin-phenol [HDBP], 50:50 ratio mixture) on cells expressing mitochondrial targeting sequence (MTS)-APEX2. (d) Scatter plot showing MS1 intensity of both LDBP- and HDBP-modified proteins labeled using MTS-APEX2 with equimolar pre-mixed probe solutions. R-squared (R²) values and trendline slopes were used to validate accuracy. (e) Experimental procedure of duplexed quantification using the ICAX method. (f) Annotated MS/MS spectra of the LDBP- and HDBP-modified peptides (PGY*AAIQALLSSR) for the PDXR protein. (g) Scatter plot showing the MS1 intensity of LDBP- and HDBP-modified peptides identified by ICAX analysis. R² values and trendline slopes in triplicate experiments are listed in the **Extended Data Fig. 3a**.

Next, we prepared either LDBP or HDBP labeled samples with cell lines stably expressing MTS-APEX2 that underwent independent labeling reactions, followed by mixing of the lysates; then we trypsinized them prior to peptide-level enrichment. This design allowed us to assess the technical variability and labeling consistency across replicates (Figure 1e). We confirmed that LDBP- and

HDBP-modified peptides were well retrieved in the LC-MS/MS spectrum (**Figure 1f**). The resulting scatter plots of log₂-transformed biotinylated peptide intensities demonstrated excellent correlation between replicates (**Figure 1g**), indicating minimal technical variation between light and heavy channels. Our findings confirm that the ICAX method effectively controls spectrum-to-spectrum variation and supports the robustness of the approach for quantitative proximity profiling.

We have updated these results including the LC-MS based comparison of labeling efficiency in the revised **Figure 1** and main text.

Figure R3. The scatter plot shows MS1 intensity of LDBP- and HDBP-modified peptides labeled by MTS-APEX2 following treatment of equimolar pre-mixed solution of LDBP and HDBP. R-squared (R^2) values and slopes of trendlines are displayed for validation of accuracy and reproducibility across triplicate experiments.

2. (Reviewer 3-2) In extended data figure 2e, the confocal images using streptavidin staining showed diffused fluorescence signal instead of overlap with anti-v5 signal. Is it because of the comparatively low signal intensity? I wonder the comparison of the labeling efficiency of the probes used in this paper with normally used APEX probes.

Response: We thank the Reviewer for raising the issue about the labeling properties of the APEX method. In previous studies addressing the mitochondrial proteome using APEX2, researchers also reported that OMM-APEX2 and IMS-APEX2 showed a diffuse pattern of biotinylated proteins with a biotin-phenol probe²²⁻²⁴. Because OMM-APEX2 faces the cytosol and hundreds of mitochondria are distributed in the cytoplasm, many cytosolic proteins are labeled by OMM-APEX2.

Figure R4. Schematic representation of the comparison of DBP labeling among OMM-, IMS-, and MTS-APEX2.

In addition, many transmembrane proteins (porins) on the outer membrane of mitochondria allow the exchange of various metabolites for mitochondrial metabolism (**Figure R4**). Thus, the desthiobiotin (or biotin) phenoxyl radical generated by IMS-APEX2 can diffuse into the cytosol resulting in the biotinylation of cytosolic proteins. The pattern of biotinylation by OMM-APEX2 showed a more cytosolic pattern compared to that of IMS-APEX2 because it generates more phenoxyl radicals in the cytosol (**Extended Data Figure 5e**). By contrast, the inner mitochondrial membrane (IMM) is generally impermeable due to its highly enriched cardiolipin and specific transport systems to maintain the proton gradient for ATP synthesis (**Figure R4**). Therefore, DBP radicals cannot passively diffuse to IMM, and consequently, MTS-APEX2 showed a restricted biotinylation pattern (**Extended Data Figure 2c**).

Extended Data Figure 5. Preparation of ascorbate peroxidase (APEX) constructs (TDRKH and SCO1) for primary intracristal subdomain/outer intracristal subdomain proteome identification. (e) Confocal images of TDRKH (1–561 aa)-APEX2 and SCO1-APEX2 stable cell lines. Biotinylated proteins were detected using streptavidin-AF568 staining. The expression of each construct was visualized using an anti-V5 antibody. The scale bars in confocal images represent 10 μ m.

Extended Data Figure 2. (c) Imaging analysis of MTS-APEX2 shows their localization and biotinylation pattern. Scale bars represent 10 μ m.

Figure R5. Comparison labeling intensity between BP and DBP.

For the Reviewer's information, the LBP used in this study is the most widely used for APEX probes. However, we could not compare the labeling efficiency between BP and DBP with western blotting, as biotin ($K_d \sim 10^{-15} \text{ M}^{-1}$) exhibits a higher affinity for streptavidin-HRP than desthiobiotin ($K_d \sim 10^{-13} \text{ M}^{-1}$)²⁵. Conversely, DBP-modified peptides can be more efficiently eluted from streptavidin-conjugated beads by heating during peptide-level enrichment²⁶. Accordingly, we found that DBP-labeled proteins exhibited slightly higher MS intensities (Figure R5, data from Extended Data Fig. 4e and Figure 1d, respectively), consistent with previous reported²⁶.

Overall, the diffusive labeling observed with OMM- and IMS-APEX2 is a characteristic feature of APEX2.

3. (Reviewer 3-3) After the author demonstrated TMEM177 and AGK was ICS-localized and OCS-localized respectively, they aimed to reveal more ICS and OCS proteomes using ICAX on TMEM177-APEX2 and AGK-APEX2 stable cell lines. Since they already identified ICS and OCS markers, is it possible to apply existing proximity labeling methods of short labeling radius (few nanometers) instead of ICAX. Compared with these proximity labeling methods of short labeling radius, what are the advantages of ICAX? Could the authors give the data of proximity labeling methods of short labeling radius and compare it with their ICAX method?

Response: We appreciate this valuable comment. Recently, proximity labeling enzymes such as APEX2 and TurboID, as well as Micromap, which is based on iridium photocatalysts, have been widely used for spatial proteome and interactome mapping. APEX2 has a comprehensive labeling radius of 200 nm, which is longer than TurboID (10–35 nm) and Micromap (10 nm)²⁷⁻³⁰. Therefore, APEX2 properties facilitate mapping sub-organelle levels of proteome including mitochondrial intermembrane space, whereas the short labeling radii of TurboID and Micromap are more suitable for interactome identification.

Extended Data Figure 1. TurboID exhibits high reactivity with endogenous biotin levels. (a) Imaging analysis of TurboID-NES and APEX2-NES. Proximity labeling enzymes localize in the cytosol. (b) Western blotting of TurboID-NES and APEX2-NES shows the biotinylation level after treatment with DMSO and 500 μ M biotin. TurboID and APEX2 levels were validated using anti-V5 antibody. GAPDH was used for loading

control. (c) LC-MS/MS analysis of biotin-modified peptides after peptide-level enrichment. Heat map of identified proteins biotinylated by TurboID-NES based on their MS intensity. (d) Venn diagram comparing the number of proteins biotinylated by TurboID-NES between DMSO-treated and excess biotin-treated samples.

Furthermore, Micromap utilizes aryl-diazirine to generate reactive carbene, via an iridium catalyst. This moiety can react with various amino acids by means of insertion into a C-H bond³¹, whereas biotin-phenol generated by APEX2 showed comparatively specific reactivity toward tyrosine rather than other electron rich amino acids³². Therefore, carbene labeling can result in numerous possible modifications on a single protein, potentially leading to low MS signal intensity of each modified peptide.

TurboID is another possibility for quantitative analysis using biotin and heavy biotin³³. Thus, we generated cell lines stably expressing APEX2-NES and TurboID-NES for comparison. We confirmed that both cell lines were well targeted in the cytosol (**Extended Data Fig 1a**); however, we found that TurboID exhibits high reactivity to endogenous biotin originally from the FBS in the cell culture media. The biotinylation level by endogenous biotin was comparable with that of treatment with excess biotin, whereas APEX2-NES exhibited no biotinylation signal in the absence of DBP (**Extended Data Fig 1b**). Then, we identified the number of biotinylated proteins by TurboID-NES with endogenous biotin using LC-MS/MS analysis after peptide-level enrichment. As a result, we found that 585 proteins were biotinylated by TurboID-NES after 500 μ M biotin and 47.2% (276 proteins) of these proteins were also labeled by TurboID-NES with endogenous biotin (**Extended Data Fig 1c, d**). Thus, the intensity of biotin-modified peptides should be higher than that of heavy biotin-modified peptides using both probes for quantitative analysis. Because biotin is essential for cell growth, it is difficult to use biotin-deficient media. Therefore, heavy biotin cannot be used for quantitative analysis either, owing to pre-blocking of lysine residues in proteins by TurboID with endogenous biotin. These results indicate that TurboID is not appropriate for our approach. Thus, we selected APEX2 for mapping the ICS and OCS in this study. We have updated all these contents in the revised manuscript.

4. (Reviewer 3-4) The authors demonstrated that their ICAX methods could identify ICS proteins and finally identified 4 proteins whose $\log_2 \text{FC}_{\text{SCO1/TDRKH}} > 20$ as strong ICS protein candidates. However, they only presented the EM images of one protein. Could the authors give the EM images of the other three proteins so as to justify the reliability of their ICAX approach?

Response: We thank the Reviewer for this valuable suggestion. Among the four identified proteins (TMEM177, NDUFA3, SLC35A4 isoform2, and APOO) using TDRKH-APEX2 and SCO1-APEX2, the APOO protein is a subunit of the MICOS complex, which localizes in the cristae junction at the entrance of ICS. Although APOO is not the innermost protein of ICS, it can be strongly labeled by SCO1-APEX2 because SCO1 localizes not only in the ICS but also in the OCS.

Therefore, we used ICS-specific TMEM177-APEX2 for mapping the ICS/OCS proteome by comparison with AGK-APEX2, which is an OCS protein in this study.

We generated NDUFA3-APEX2 and SLC35A4 isoform2-APEX2 stable cell lines in Flp-In T-Rex 293 cells for APEX-EM experiment. However, we found that the V5 and TOM20 signals were not matched, indicating that localization of those proteins was inhibited by APEX2 conjugation (**Figure R5**) owing to their small size (NDUFA3: 84 amino acids, SLC35A4 isoform 2: 103 amino acids). These results were confirmed in HeLa cells. Although we are unable to provide EM images, we believe that NDUFA3 localizes in the ICS, as it is a subunit of OXPHOS complex I, and it was identified in ICS in our TMEM177-APEX2/AGK-APEX2 dataset.

Figure R5. Confocal images of Flp-In T-Rex 293 and HeLa cells expressing NDUFA3- and SLC35A4 isoform2-APEX2, respectively. The expression pattern of APEX2 was visualized by anti-V5 antibody and the biotinylated proteins were stained with streptavidin-AF568 or AF647. Mitochondria were visualized by anti-TOM20 antibody. Scale bar = 10 μ m.

5. (Reviewer 3-5) The authors classified the ICAX-labeled proteins in the experiment of SCO1-APEX2 and TDRKH-APEX2 into 4 sub-mitochondrial populations based on their FC_SCO1/TDRKH values. They chose proteins of \log_2 FC_SCO1/TDRKH > 20 as group 1. Are

there any criteria for the chosen of FC value? Why chose 20? In Line 252, the author mentioned that “we selected 64 proteins that were highly enriched by TMEM177-APEX2 (group ICS)”. There seems to be no mention of the screening criteria.

Response: We apologize for the insufficient explanation. The purpose of ICAX analysis using SCO1- and TDRKH-APEX2 stable cells is identification of the ICS-specific protein which has not been reported yet. In the SCO1-APEX2/TDRKH-APEX2 dataset, we found that 4 proteins (TMEM177, NDUFA3, SLC35A4 isoform2, APOO) were exclusively labeled by SCO1-APEX2, indicating that those proteins were not labeled by TDRKH-APEX2 (**Figure R6a**). Therefore, we selected these 4 proteins as group 1 which has high potential to be specifically localized in the ICS.

Figure R9. (a) Bar graph showed Log₂ transformed MS1 intensity of 4 identified proteins (TMEM177, NDUFA3, SLC35A4 isoform2, APOO) in Group I and AGK protein in Group II. (b) Proposed localization of the MICOS complex and OPA1 in the mitochondria.

Since SCO1 is not ICS-specific protein, the SCO1-APEX2 can also label OCS proteins. Hence, we performed further ICAX analysis using TMEM177- and AGK-APEX2 stable cells to achieve more accurate mapping of ICS and OCS proteome. Based on our ICAX analysis, we can estimate the relative distance of labeled proteins from APEX2 conjugated proteins. However, the labeling efficiency of TMEM177- and AGK-APEX2 is not exactly the same due to different expression levels and localization. It indicates that we cannot directly define that proteins labeled by TMEM177-APEX2 are ICS proteins. Thus, we referred to the fold change of the OPA1 (Log₂ FC: 0.71) and YME1L1 (Log₂ FC: 0.62) which may localize the outermost part of ICS space for regulation of cristae structures (**Figure R6b**). MICOS complexes also localize in cristae junction, however it may face the OCS for interaction with several complexes localized at OMM.

We have described the criteria for selecting 64 proteins for the ICS group in the revised manuscript accordingly.

6. (Reviewer 3-6) The author used the TMEM177-APEX2 stable cell line to monitor the changes in ICS proteome under conditions of energy stress. However, if TMEM177 would relocalize, the above conclusions would be unreliable. The authors should present some experimental evidence that TMEM177 would not relocalize under conditions of energy stress.

Response: Thank you for your insightful comment. As the reviewer suggested, we conducted several experiments. First, we express the TMEM177-roGFP in the Flp-In T-Rex 293 cells stably expressing MTS-APEX2 to examine whether topology of TMEM177 was changed during mitochondrial uncoupling process. We found that biotinylated TMEM177-roGFP was not changed by BAM15 treatment (**Extended Data Figure 14a**), indicating that the topology of TMEM177 remained. We also conducted APEX-EM imaging using stable cell lines expressing TMEM177-APEX2 for checking TMEM177 remained in the ICS after BAM15 treatment. As a result, we found that the DAB staining signal of APEX exhibited ICS pattern rather than OCS pattern in both BAM15-treated cells and DMSO-treated cells (**Extended Data Figure 14b**). This result indicates that the localization of TMEM177-APEX2 in ICS was not altered by BAM15 treatment.

Extended Data Figure 14. Validation of mitochondrial matrix proteins labeled with TMEM177-APEX2 under BAM15 treatment. (a) Proximity biotinylation of MTS-APEX2 for TMEM177-conjugated fluorescent protein (roGFP) with or without BAM15 treatment. Biotinylated TMEM177-roGFP was detected via western blotting with an anti-GFP antibody. The labeling of mitochondrial matrix proteins by MTS-APEX2 was confirmed through immunoblotting with anti-HSPD1 (mitochondrial matrix) and anti-TOM20 (outer mitochondrial membrane) antibodies. The proximal biotinylation level of TMEM177-roGFP labeled with MTS-APEX2 remained unchanged following BAM15 treatment. (b) Electron microscopic images of TMEM177-APEX2 after diaminobenzidine staining under mitochondrial uncoupling.

7. (Reviewer 3-7) The discussion section was too brief. Could the authors give more perspective and applications of this ICAX method.

Response: We appreciate this comment. As the reviewer suggested, we have updated Discussion section with our perspective and applications.

8. (Reviewer 3-8) There might exist some errors in this manuscript. For example, “Fig. 1g” in Line 197 probably be “Fig. 2g” and “Fig. 2f” in Line 213 probably be “Fig, 2g”. “pHluorin2” in the last line of images in extended data figure 8a probably be “gTEMP”. The authors need to be further checked.

Response: We apologize for these errors. We have updated our manuscript accordingly.

Reference

1. Park, I. *et al.* Mitochondrial matrix RTN4IP1/OPA10 is an oxidoreductase for coenzyme Q synthesis. *Nat Chem Biol* **20**, 221-233 (2024).
2. Dumrongprechachan, V. *et al.* Cell-type and subcellular compartment-specific APEX2 proximity labeling reveals activity-dependent nuclear proteome dynamics in the striatum. *Nat Commun* **12**, 4855 (2021).
3. Savitski, M.M. *et al.* Measuring and managing ratio compression for accurate iTRAQ/TMT quantification. *J Proteome Res* **12**, 3586-3598 (2013).
4. Shin, S. *et al.* Super-resolution proximity labeling with enhanced direct identification of biotinylation sites. *Commun Biol* **7**, 554 (2024).
5. Zecha, J. *et al.* TMT Labeling for the Masses: A Robust and Cost-efficient, In-solution Labeling Approach. *Mol Cell Proteomics* **18**, 1468-1478 (2019).
6. Rath, S. *et al.* MitoCarta3.0: an updated mitochondrial proteome now with sub-organelle localization and pathway annotations. *Nucleic Acids Research* **49**, D1541-D1547 (2021).
7. Sekine, S. *et al.* Rhomboid protease PARL mediates the mitochondrial membrane potential loss-induced cleavage of PGAM5. *J Biol Chem* **287**, 34635-34645 (2012).
8. Le Vasseur, M. *et al.* Genome-wide CRISPRi screening identifies OCIAD1 as a prohibitin client and regulatory determinant of mitochondrial Complex III assembly in human cells. *Elife* **10** (2021).
9. UniProt, C. UniProt: the Universal Protein Knowledgebase in 2025. *Nucleic Acids Res* **53**, D609-D617 (2025).
10. Hallgren, J. *et al.* (2022).
11. Bernhofer, M. & Rost, B. TMbed: transmembrane proteins predicted through language model embeddings. *BMC Bioinformatics* **23** (2022).
12. Abramson, J. *et al.* Accurate structure prediction of biomolecular interactions with AlphaFold 3. *Nature* **630**, 493-500 (2024).
13. Stephan, T. *et al.* MICOS assembly controls mitochondrial inner membrane remodeling and crista junction redistribution to mediate cristae formation. *EMBO J* **39**, e104105 (2020).
14. Lee, Y.B. *et al.* Super-resolution proximity labeling reveals anti-viral protein network and its structural changes against SARS-CoV-2 viral proteins. *Cell Rep* **42**, 112835 (2023).
15. Spiller, M.P., Guo, L., Wang, Q., Tran, P. & Lu, H. Mitochondrial Tim9 protects Tim10 from degradation by the protease Yme1. *Biosci Rep* **35** (2015).
16. Rockfield, S.M. *et al.* Genetic ablation of Immt induces a lethal disruption of the MICOS complex. *Life Sci Alliance* **7** (2024).
17. Luis-Garcia, E.R. *et al.* Mitochondrial Dysfunction and Alterations in Mitochondrial Permeability Transition Pore (mPTP) Contribute to Apoptosis Resistance in Idiopathic Pulmonary Fibrosis Fibroblasts. *Int J Mol Sci* **22** (2021).
18. Zhao, Z. *et al.* Modulation of intracellular calcium waves and triggered activities by mitochondrial ca flux in mouse cardiomyocytes. *PLoS One* **8**, e80574 (2013).
19. NavaneethaKrishnan, S., Rosales, J.L. & Lee, K.Y. mPTP opening caused by Cdk5 loss is due to increased mitochondrial Ca(2+) uptake. *Oncogene* **39**, 2797-2806 (2020).
20. Pak, V.V. *et al.* Ultrasensitive Genetically Encoded Indicator for Hydrogen Peroxide Identifies Roles for the Oxidant in Cell Migration and Mitochondrial Function. *Cell Metab* **31**, 642-653 e646 (2020).

21. Dickson-Murray, E., Nedara, K., Modjtahedi, N. & Tokatlidis, K. The Mia40/CHCHD4 Oxidative Folding System: Redox Regulation and Signaling in the Mitochondrial Intermembrane Space. *Antioxidants (Basel)* **10** (2021).
22. Hung, V. *et al.* Proteomic mapping of the human mitochondrial intermembrane space in live cells via ratiometric APEX tagging. *Mol Cell* **55**, 332-341 (2014).
23. Lee, S.Y. *et al.* APEX Fingerprinting Reveals the Subcellular Localization of Proteins of Interest. *Cell Rep* **15**, 1837-1847 (2016).
24. Hung, V. *et al.* Proteomic mapping of cytosol-facing outer mitochondrial and ER membranes in living human cells by proximity biotinylation. *Elife* **6** (2017).
25. Levy, M. & Ellington, A.D. Directed evolution of streptavidin variants using in vitro compartmentalization. *Chem Biol* **15**, 979-989 (2008).
26. Lee, S.Y. *et al.* Architecture Mapping of the Inner Mitochondrial Membrane Proteome by Chemical Tools in Live Cells. *J Am Chem Soc* **139**, 3651-3662 (2017).
27. Reeves, A.E. & Huang, M.L. Proximity labeling technologies to illuminate glycan-protein interactions. *Curr Opin Chem Biol* **72**, 102233 (2023).
28. Oakley, J.V. *et al.* Radius measurement via super-resolution microscopy enables the development of a variable radii proximity labeling platform. *Proc Natl Acad Sci USA* **119**, e2203027119 (2022).
29. Kreis, E. *et al.* TurboID reveals the proximiomes of Chlamydomonas proteins involved in thylakoid biogenesis and stress response. *Plant Physiol* **193**, 1772-1796 (2023).
30. Kang, M.G. & Rhee, H.W. Molecular Spatiomics by Proximity Labeling. *Acc Chem Res* **55**, 1411-1422 (2022).
31. West, A.V. *et al.* Labeling Preferences of Diazirines with Protein Biomolecules. *J Am Chem Soc* **143**, 6691-6700 (2021).
32. Udeshi, N.D. *et al.* Antibodies to biotin enable large-scale detection of biotinylation sites on proteins. *Nat Methods* **14**, 1167-1170 (2017).
33. Stockhammer, A. *et al.* When less is more - a fast TurboID knock-in approach for high-sensitivity endogenous interactome mapping. *J Cell Sci* **137** (2024).

REVIEWERS' COMMENTS

Reviewer #1 (Remarks to the Author):

The authors have addressed my previous comments clearly and convincingly. The investigations on TMT labeling of biotinylated and attempted optimization are excellent and reinforce the value of the approach. The new figures on the topology of biotinylated residues add significantly to the confidence on the specificity of labeling. The manuscript is well written and clearly presented. I congratulate the authors on an excellent study have no problem recommending publication.

We sincerely appreciate the Reviewer's kind and thoughtful comments. We are especially grateful for the insightful and constructive suggestions regarding the concept of ICAX analysis, which enabled us to more effectively emphasize the advantages of the tool throughout the revision process. In addition, the Reviewer's concern regarding ambiguous scientific terms described in the previous manuscript helped us improve its readability. We believe that these comments have significantly contributed to enhancing the overall quality and clarity of the revised manuscript.

Reviewer #2 (Remarks to the Author):

The authors now added a large number of additional data and the MIC60 depletion dataset revealed many novel interesting aspects. This is an excellent study of very high technical quality. The tools developed here will be valuable for many researchers. It is interesting that the authors observed that mitoribosomal proteins can accumulate in the intermembrane space in human cells. This is in agreement with two yeast studies which recently were published and the authors might cite them in the context of matrix proteins that were found in the intermembrane space (PMID: 39890954, 40524011).

I now fully support the publication of this study in its present form.

Response: Thank you for this valuable suggestion. We have cited the references (PMID: 39890954, 40524011) with the following description: "Recent studies have revealed that mitochondrial matrix proteins, including ribosomal subunits, are accumulated in the IMS of compromised mitochondria upon inhibition of ATP synthase in *yeast*, potentially preventing their mistargeting to the nucleus."

We sincerely appreciate the Reviewer's kind and positive feedback. The Reviewer's helpful and constructive suggestions have guided us in validating our ICS proteome analysis under both normal and dynamic stress conditions throughout the revision process. We believe that these thoughtful comments have greatly contributed to improving the overall quality and clarity of the revised manuscript.

Reviewer #3 (Remarks to the Author):

The authors have addressed all of my previous comments and questions. I have no objection to publication.

We sincerely appreciate the Reviewer's positive evaluation. The Reviewer's insightful and valuable suggestions regarding the comparison of our APEX-based ICAX analysis with other proximity labeling tools were extremely helpful in clarifying the rationale for selecting APEX. Additionally, the comments concerning the criteria applied to our ICS proteome mapping results have greatly contributed to enhancing the overall quality and readability of the revised manuscript.